# Chain of Thought with Explicit Evidence Reasoning for Few-shot Relation Extraction

**Xilai Ma, Jing Li**[*] and **Min Zhang**
Harbin Institute of Technology, Shenzhen, China
`maxilai.hour1@gmail.com`
`{li.jing, zhangmin2021}@hit.edu.cn`

## Abstract

Few-shot relation extraction involves identifying the type of relationship between two specific entities within a text, using a limited number of annotated samples. A variety of solutions to this problem have emerged by applying meta-learning and neural graph techniques which typically necessitate a training process for adaptation. Recently, the strategy of in-context learning has been demonstrating notable results without training. Few studies have already utilized in-context learning for zero-shot information extraction. Unfortunately, the evidence for inference is either not considered or implicitly modeled during the construction of chain-of-thought prompts. In this paper, we propose a novel approach for few-shot relation extraction using large language models, named CoT-ER, chain-of-thought with explicit evidence reasoning. In particular, CoT-ER first induces large language models to generate evidence using task-specific and concept-level knowledge. Then this evidence is explicitly incorporated into chain-of-thought prompting for relation extraction. Experimental results demonstrate that our CoT-ER approach (with 0% training data) achieves competitive performance compared to the fully-supervised (with 100% training data) state-of-the-art approach on the FewRel1.0 and FewRel2.0 datasets.

## 1 Introduction

Relation extraction (RE) aims at identifying the relation between two given entities based on contextual semantic information (Cardie, 1997; Bach and Badaskar, 2007; Pawar et al., 2017). However, the performance of RE models often degrades significantly when the labeled data is insufficient. The few-shot relation extraction (FSRE) task needs to be addressed with a limited amount of annotated training data (Han et al., 2018; Gao et al., 2019b; Brody et al., 2021). Recently, numerous

---

[*] Corresponding author.

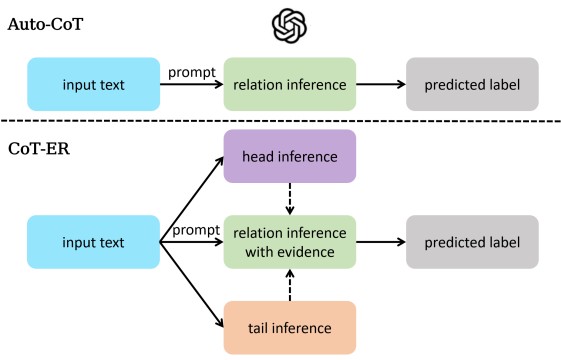

Figure 1: The comparison between Auto-CoT and CoT-ER (ours) prompting methods. Specifically, CoT-ER leverages side information to induce LLMs to generate explicit evidence for relation reasoning.

researchers have tackled this problem by employing meta-learning and neural graph techniques (Fangchao et al., 2021; Dou et al., 2022; Li and Qian, 2022; Zhang et al., 2021; Li et al., 2023). These methods have achieved satisfying results by meta-training the model on a large dataset or incorporating external knowledge.

More recently, pre-trained Large Language Models (LLMs) such as GPT-series models, have exhibited significant in-context learning capabilities (Brown et al., 2020; Min et al., 2022), achieving promising results across many NLP tasks. These findings suggest that LLMs can effectively perform various tasks without the need for parameter optimization, a concept known as In-context Learning (Dong et al., 2022). Within the paradigm of in-context learning, LLMs demonstrate competitive performance compared to standard fully-supervised methods across many NLP tasks, even with just a few examples provided as few-shot demonstrations in the prompt (Wang et al., 2022).

Furthermore, the chain-of-thought prompting method (Wei et al., 2022; Kojima et al., 2022; Zhang et al., 2022) elicits an impressive reasoning capability from the LLM in mathematical problems and commonsense reasoning. While in the

RE task, there may exist a reasoning process that guides the LLM in determining the relation label. However, there is a lack of research to fill this gap. Though GPT-RE (Wan et al., 2023) introduces a golden label-induced reasoning method by prompting the LLM to generate a suitable reasoning process solely based on the given golden label. The performance improvement from the auto-generated reasoning process is marginal compared to a meticulously designed approach for few-shot demonstration retrieval.

We argue that the one-step reasoning process generated by LLM does not fully unleash the potential of LLM: (1) Previous studies and our experiments indicate that the one-step auto-generated reasoning process by LLM does not emphasize the higher-level abstraction of entity types, specifically the concept-level entities, which has been proven to be beneficial for FSRE task (Hao et al., 2019; Zhang et al., 2021). For instance, consider the following simple example: the relation between two location entities should not be categorized under the relation type between human beings. (2) Due to the huge amount of pre-training data, the LLM has already possessed a considerable knowledge base (Petroni et al., 2019; Roberts et al., 2020), which can be beneficial when the LLM encounters an FSRE task. (3) The quality of semantic representation of the relation label is not crucial in the fully-supervised setting, but in-context learning is sensitive to the relation label. For instance, given the relation labels= $\{mother, child, sport\}$ in FewRel 1.0 (Han et al., 2018), relation labels "$mother$" and "$child$" would confuse the LLM without appropriate prompt designing, as the primary distinction between these two relations is the positioning of the parent entity as either the head or tail entity. Moreover, the word "$sport$" barely contains enough relation information for the LLM to perform the RE task. We call this issue the semantic ambiguity of relation labels.

To this end, this paper presents a novel chain-of-thought prompting method for the FSRE task: Chain-of-thought with Explicit Evidence Reasoning, achieving competitive results compared to the state-of-the-art result on FewRel 1.0 and FewRel 2.0. Our method employs a 3-step reasoning approach to address the aforementioned issues. In the first and second steps, CoT-ER requires the LLM to output the concept-level entities corresponding to the head and tail entities, which serve as the

foundation for RE-specific reasoning. In the third step, CoT-ER prompts the LLM to extract the relevant contextual spans as evidence that explicitly establishes a specific relationship between these two entities. By combining the head entity, tail entity, and relation label to form a coherent sentence, LLMs can determine the relation label between two given entities more semantically, addressing the issue of semantic ambiguity of relation labels in prompting methods. Figure 1 demonstrates the difference between Auto-CoT and our CoT-ER.

## 2 Related Work

**Few-shot Relation Extraction.** Few-shot relation extraction aims at predicting semantic relations between head and tail entities indicated in a given instance based on a limited amount of annotated data. FewRel, a large-scale dataset introduced by Han et al. (2018), was the first to explore few-shot learning in relation extraction. Many approaches (Qu et al., 2020; Yang et al., 2021; Zhang et al., 2021) incorporate external knowledge to improve performance given the scarcity of training data. Another line of FSRE research (Gao et al., 2019a; Han et al., 2021; Liu et al., 2022b) relies solely on the input text and provided relation description information, without incorporating external knowledge. Most of the previous methods usually adopt complicated designs of neural networks or introduce external knowledge, which can be labor-intensive in realistic scenarios.

**In-context Learning.** GPT-3 in-context learning (ICL) (Brown et al., 2020; Dong et al., 2022) has emerged as a novel paradigm in NLP and demonstrates competitive performance across various tasks when compared to fine-tuned models. It's much easier to introduce prior knowledge into LLMs by incorporating relevant text information into prompt (Liu et al., 2022a; Lu et al., 2022; Wei et al., 2022). Furthermore, ICL is a training-free approach by directly prompting the LLMs, which means it's a ready-to-use method and can be easily applied to various tasks with a few demonstrations in the prompt.

Recently, most researchers focus on the demonstration designing of ICL to improve the performance in NLP tasks and gradually developed into two categories (Dong et al., 2022). The first line of demonstration designing tries to seek an optimal arrangement of the few-shot demonstrations in the

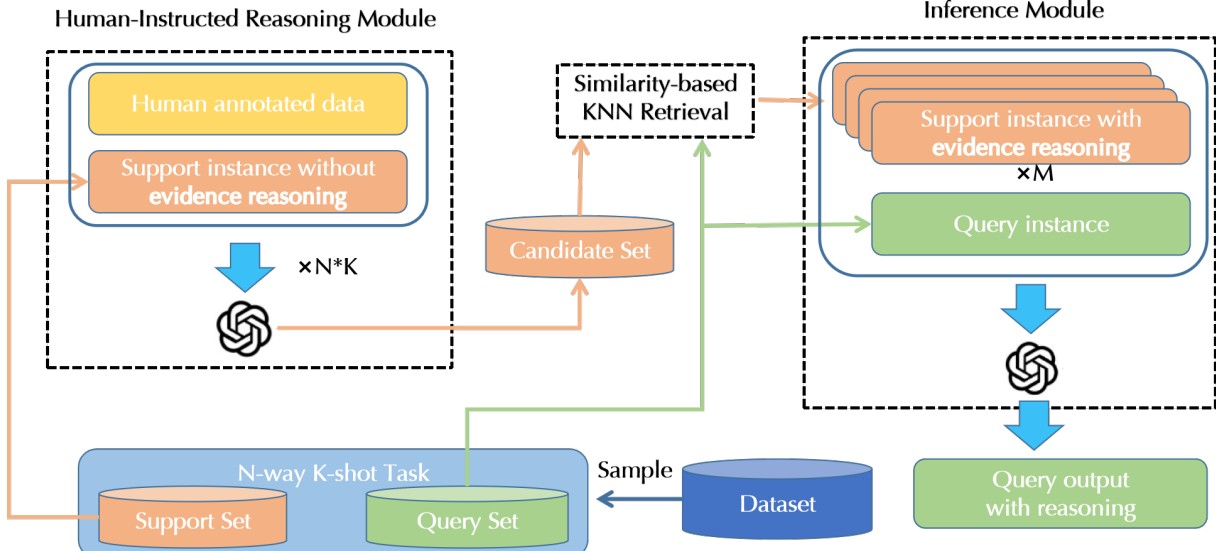

Figure 2: An illustration of CoT-ER for few-shot RE. Different colored lines indicate the flow of support and query instances from an N-way K-shot task. a) Human-instructed reasoning module (§3.3) associates an evidence reasoning process with each instance from the support set by prompting LLM with human-annotated demonstrations; b) Instances retrieval module (§3.4) selects the few-shot demonstrations from the candidate set for the ultimate prompt based on their similarity to the query instance. c) Inference module (§3.5) utilizes the ultimate prompt, which is composed of $\mathcal{M}$ support instances with their associated reasoning process, to derive an evidence reasoning process for the query instance.

prompt by selecting instances from the dataset (Liu et al., 2022a; Valmeekam et al., 2022; Wu et al., 2022; Wan et al., 2023) and ordering the selected demonstration examples (Lu et al., 2022; Liu et al., 2022a; Ma et al., 2023). Another line of demonstration design aims to discover an effective prompting method to unleash the potential of LLMs. Several studies (Honovich et al., 2022; Zhou et al., 2022) find that the LLMs can generate the task instruction automatically. Furthermore, Wei et al. (2022) revealed the reasoning ability of LLM by adding intermediate reasoning steps manually before giving the answer, which is called chain-of-thought (CoT). Additionally, Kojima et al. (2022) shows that by simply adding *"Let's think step by step"* before each answer, LLM can do the zero-shot reasoning without manually annotated data. Based on this discovery, Zhang et al. (2022) proposed Auto-CoT, replacing the manually written reasoning process in CoT with the automatically generated reasoning process by LLM.

Despite the CoT prompting method achieving promising results in many NLP tasks, it still lacks relevant exploration for RE. Hence, in this paper, we propose a novel CoT prompting method called CoT-ER to fill this gap.

**True Few-Shot Learning.** Perez et al. (2021) argues that prior research has achieved promising results by choosing prompts or tuning other hyperparameters using a large development set, such as selecting few-shot demonstrations from a large training set, which does not truly demonstrate the few-shot learning capability of LLMs. This setting has been adopted by many works (Logan IV et al., 2022; Schick and Schütze, 2022; Lu et al., 2022; Jimenez Gutierrez et al., 2022) to get a more accurate result of few-shot performance. We will also adopt the setting in this paper.

## 3 CoT-ER

### 3.1 Problem Formulation

**Definition of Relation Extraction.** Let $x$ denote the input sentence and $e_{sub}$, $e_{obj}$ denote the pair of subject and object entities in the given sentence. The RE task aims to identify the relation label $r$ between the marked entities in sentence $x$. Here $\mathcal{R}$ represents a predefined set of relations, and $r$ is an element of $\mathcal{R}$.

**Definition of Few-shot Relation Extraction.** Given an N-way K-shot RE task, the goal is to solve this problem for each instance in the query set based on the support set. The relation label

set $\mathcal{R}$ consists of $N$ type of relations. For each $r \in \mathcal{R}$, the support set $\mathcal{S}_r$ includes $K$ instances, represented as $\{s_r^1, s_r^2, s_r^3, ..., s_r^K\}$. The query set $Q$ comprises the test input instance for each $r \in \mathcal{R}$.

Since the $N$ and $K$ are usually quite small, predicting relations in query instances with limited labeled data presents a significant challenge. Previous studies tackled this problem by training a data-efficient network, specifically the meta-learning-based method. In the subsequent section, we will discuss a training-free method to address this problem by leveraging the reasoning ability of the LLM.

### 3.2 Overview

An overview of our proposed CoT-ER is shown in Figure 2, which consists of 3 components: (1) The **Human-Instructed Reasoning Module**, which aims to associate a reasoning process with each instance from the support set by prompting LLM with human-annotated data. (2) A **Similarity Based KNN Retrieval Module** will select instances with the reasoning process from the support set based on the similarity to query instance, which are considered as few-shot demonstrations in the ultimate prompt. (3) The **Inference Module** predicts the relation label of a query instance by instructing the LLM through the ultimate prompt, which concatenates the task instruction, few-shot demonstrations, and a question about the instance.

### 3.3 Human-Instructed Reasoning Module

Since the LLM has the ability of in-context learning (Brown et al., 2020), we propose a human-instructed approach to guide the LLM in performing accurate reasoning using a minimal amount of annotated data.

**CoT-ER Designing.** To fully leverage the knowledge stored in LLM and facilitate step-by-step reasoning, we introduce a novel 3-step reasoning framework with concept-level knowledge and explicit evidence. In **Step 1**, the LLM infers concept-level knowledge related to the head entity, while **Step 2** does the same for the tail entity. Through these steps, the LLM can easily exclude options with incorrect concept entities. **Step 3**: To figure out which relation label fits this pair of entities most within the given context, we explicitly highlight relevant text spans as evidence, and subsequently construct a coherent expression that combines the two entities and the relation label together. Table 6 shows an example with the

relation label "crosses". To further illustrate our 3-step reasoning process, few-shot demonstrations in Figure 3 demonstrate the template of this reasoning process.

**CoT-ER Generating.** We annotated one CoT-ER reasoning example for each relation class in the dataset to be seed examples.[1] Then we design an appropriate prompt[2] using the annotated example as the few-shot demonstration to instruct the LLM in generating a similar reasoning step for each support instance. Each support instance with the CoT-ER reasoning steps will be appended to the candidate set. Figure 3 shows a similar prompt designed for the Human Instructed Reasoning Module.

### 3.4 Instance Retrieval Module

Several studies (Liu et al., 2022a; Lu et al., 2022; Wan et al., 2023) suggest that selecting few-shot demonstrations based on similarity yields strong improvements in in-context learning. Wan et al. (2023) achieved promising performance in RE by employing a task-specific fine-tuned model as the encoder, which means this approach does not fit the true few-shot setting mentioned in §2. Moreover, this advantage diminishes rapidly as the number of candidates decreases.

Because of the limited input tokens of LLM, a single prompt may not hold all support instances given an N-Way K-Shot task. For instance, in the case of the 10-way 5-shot task, having a total of 50 candidate samples leads to the inability to append them all in one single prompt. In this paper, we follow the similarity-based method for selecting few-shot demonstrations. To obtain a relation-specific similarity representation, we first reconstruct the input text as "Context: [text] Given the context, what is the relation between "[head entity]" and "[tail entity]"?" by incorporating entity-level information. Then, we utilize the GPT series model "text-embedding-ada-002" as the encoder to get the semantic embedding. Subsequently, we compute the Euclidean distance between each instance in the candidate set and the query instance. Finally, $\mathcal{M}$ instances from the candidate set are selected as few-shot demonstrations based on their lower Euclidean distance to the query instance. Intuitively, we aim to provide as much information as possible for LLM, so we follow the principle of filling the

---

[1]All seed examples are shown in Appendix B.
[2]All prompts are shown in Appendix C.

Figure 3: Template of the ultimate prompt. $\mathcal{M}$ represents the number of few-shot demonstrations selected by the instance retrieval module, and $Verbalize()$ denotes a transformation function that combines the component into a coherent expression. The prompt used in human human-instructed reasoning module follows a similar structure, but instead of few-shot demonstrations, it employs annotated examples.

context window to increase $\mathcal{M}$ as much as possible.

## 3.5 Inference Module

To create the ultimate prompt, we simply concatenate a task instruction, few-shot demonstrations, and a question that is tailored to the query instance, using the support instances with CoT-ER reasoning as few-shot demonstrations. Figure 3 shows the framework of the ultimate prompt. It is worth noting that LLMs have a strong inclination to wrongly output $NULL$ in a general setting (Wan et al., 2023; Jimenez Gutierrez et al., 2022). Here, we enforce the LLM to select one of the provided relation labels, as we do not consider the "None-of-the-Above" scenario in the FewRel dataset (Han et al., 2018; Gao et al., 2019b).

## 4 Experimental Setups

### 4.1 Dataset For Few-shot Relation Extraction

There are two standard few-shot relation extraction datasets: FewRel 1.0 (Han et al., 2018) and FewRel 2.0 (Gao et al., 2019b)[3]. FewRel 1.0 is constructed from Wikipedia, which consists of 70,000 sentences annotated with 100 relation labels, these

---

[3] https://github.com/thunlp/FewRel

100 relation labels are divided into 64/16/20 splits for train/validation/test set. FewRel 2.0 extends FewRel 1.0 by introducing additional validation and test sets from the medical domain, which includes 10 relation labels with 1,000 instances and 15 relation labels with 1,500 instances, respectively. Besides, note that FewRel 1.0 provides a description of each relation label in the dataset, but FewRel 2.0 does not. This difference is a crucial factor to consider when designing the seed CoT-ER reasoning process[4].

### 4.2 Implementation Details

For LLM, we select "text-davinci-003" and get the response from GPT by calling the open API of the OpenAI[5] with the parameter $temperature = 0$.

In a realistic scenario, it's reasonable to perform the RE task directly with fixed, manually annotated examples as few-shot demonstrations for each relation label. To this end, we evaluate the performance by selecting the few-shot demonstrations from a predetermined human-annotated CoT-ER dataset (seed examples), which is denoted as **Manual-CoT-ER**. In this setting, the few-shot demonstrations are independent of the support set, meaning that LLM will perform the RE task using a smaller amount of annotated data. In contrast, **Auto-CoT-ER** utilizes the auto-generated CoT-ER reasoning process as the few-shot demonstrations based on the support set which is described in §3.3.

Following the standard configuration of FewRel, we conducted experiments in these settings: 5-Way 1-shot, 5-Way 5-shot, 10-Way 1-Shot, and 10-Way 5-Shot. Due to the high cost of running the CoT-ER on the GPT-3 API and the golden labels of the test set are not publicly available, we evaluate all LLM-based methods by sampling $100 \times N$ test queries for each N-way K-shot task from validation sets.

### 4.3 Compared Methods

We consider two categories of methods for the FSRE task. **Methods with** $100\%$**-training data:** MTB (Baldini Soares et al., 2019), CP (Peng et al., 2020), HCPR (Han et al., 2021), FAEA (Dou et al., 2022), GTPN (Fangchao et al., 2021), GM_GEN (Li and Qian, 2022), KEFDA (Zhang et al., 2021). Generally, these methods train a model on FewRel 1.0 training set and evaluate their performance on FewRel 1.0, 2.0 validation and test sets.

---

[4] More details are demonstrated in Appendix B.

[5] https://platform.openai.com/docs/api-reference

| Methods | 5-Way 1-Shot | 5-Way 5-Shot | 10-Way 1-Shot | 10-Way 5-Shot |
|---|---|---|---|---|
| *Fully-Supervised (100% training data)* | | | | |
| GTPN | - / 89.40 | - / 97.00 | - / 84.40 | - / 93.80 |
| CP | -/95.10 | -/97.10 | -/91.20 | -/94.70 |
| FAEA | 90.81 / 95.10 | 94.24 / 96.48 | 84.22 / 90.12 | 88.74 / 92.72 |
| HCPR | 94.10 / 96.42 | 96.05 / 97.96 | 89.13 / 93.97 | 93.10 / 96.46 |
| GM_GEN | 96.97 / 97.03 | 98.32 / 98.34 | 93.97 / 94.99 | 96.58 / 96.91 |
| *Training-free Baselines (0% training data)* | | | | |
| Bert-proto | 29.32/- | 35.18/- | 17.84/- | 23.88/- |
| GPT-proto | 63.20/- | 82.00/- | 52.10/- | 71.50/- |
| Vanilla-ICL | 96.20(5)/- | 97.00(25)/- | 89.20(10)/- | 93.90(40)/- |
| Auto-CoT | 94.60(5)/- | 95.80(20)/- | 87.40(10)/- | 91.40(20)/- |
| + reasoning | 95.40(5)/- | 96.40(15)/- | 87.60(10)/- | 92.40(15)/- |
| *CoT-ER (0% training data)* | | | | |
| Auto-CoT-ER | 97.40(5)/- | 97.00(13)/- | 92.10(10)/- | 94.70(13)/- |
| Manual-CoT-ER | 97.00(5)/- | - | 92.60(10)/- | - |

Table 1: Main Results on FewRel 1.0 validation/test set. All results are given by accuracy (%). $(m)$ means there are $m$ instances selected as few-shot demonstrations. $+reasoning$ denotes the demand of reasoning process generation.

**Methods with $0\%$-training data:** To the best of our knowledge, no relevant evaluation has been conducted under the N-Way K-Shot setting on the FSRE dataset FewRel using the in-context learning approach. Thus we applied **Vanilla-ICL** (Brown et al., 2020) and **Auto-CoT** (Zhang et al., 2022; Wan et al., 2023) as the baseline prompt formatting methods. These methods utilize a few examples as demonstrations and prompt the LLM to perform an NLP task. **Vanilla-ICL** designs a template that directly combines the texts and relation label, such as "Context:[text], Given the context, the relation between [head entity] and [tail entity] is [relation label]". **Auto-CoT** extends the Vanilla-ICL with auto-generated reasoning steps. Throughout the experiment, we noticed that whether to require the LLM to perform reasoning in the final answering stage can lead to inconsistent results, thus we report both of the results in Table 1 and Table 2. Additionally, we utilize the pre-trained BERT-base model[6] and the GPT series model text-embedding-ada-002 as the encoder to directly obtain a representation of the input text. For each N-way K-shot task, we obtain a prototype of each class by averaging $K$ instances that belong to this class. Then the predicted label of the query instance is assigned to the class whose prototype has the closest Euclidean distance to the query instance. We denote these two methods as **Bert-proto** and **GPT-proto**.

## 5 Results and Discussion

### 5.1 Main Results

We present our main experiment results with previous methods in Table 1 and Table 2. From the table, we can observe that:

First, Auto-CoT does not demonstrate significant improvement compared to Vanilla-ICL in the few-shot scenario. This could be attributed to the low quality of the reasoning process and the reduced number of instances in few-shot demonstrations due to the maximum tokens limitation. Furthermore, When it comes to generating a reasoning process in the ultimate answer, Auto-CoT with reasoning outperforms the version of directly generating a relation label on FewRel 1.0. However, an opposite conclusion is reached on FewRel 2.0. We try to provide an explanation for this: FewRel 1.0 draws instances from Wikipedia and often requires common sense for reasoning, whereas FewRel 2.0 necessitates medical-related expertise and constitutes a smaller portion of the pre-training corpus compared to common sense. Consequently, the LLM encounters difficulties in performing reasoning tasks in the medical domain.

Second, Both Manual-CoT-ER and Auto-CoT-ER outperform the training-free baselines with fewer instances used in the few-shot demonstrations. Indicating the necessity of designing a specific CoT prompting method tailored to the RE task in order to achieve better performance in the

---

[6] https://github.com/huggingface/transformers

| Methods | 5-Way 1-Shot | 5-Way 5-Shot | 10-Way 1-Shot | 10-Way 5-Shot |
|---|---|---|---|---|
| *Fully-Supervised (100% training data)* | | | | |
| MTB | -/74.70 | -/87.90 | -/62.50 | -/81.10 |
| HCPR | -/76.34 | -/83.03 | -/63.77 | -/72.94 |
| CP | -/79.70 | -/84.90 | -/68.10 | -/79.80 |
| FAEA | -/73.58 | -/90.10 | -/62.98 | -/80.51 |
| GTPN | 82.8 / 80.0 | 91.4 / 92.6 | 71.0 / 69.25 | 86.0 / 86.9 |
| KEFDA | 86.18 / 87.81 | 94.38 / 95.00 | 79.46 / 81.84 | 90.77 / 90.63 |
| *Training-free Baselines (0% training data)* | | | | |
| Bert-proto | 27.30/- | 33.22/- | 15.80/- | 20.65/- |
| GPT-proto | 54.80/- | 81.40/- | 41.90/- | 63.90/- |
| Vanilla-ICL | 83.40$_{(5)}$/- | 89.80$_{(25)}$/- | 68.00$_{(10)}$/- | 81.40$_{(30)}$/- |
| Auto-CoT | 81.34$_{(5)}$/- | 89.00$_{(15)}$/- | 70.40$_{(10)}$/- | 80.7$_{(17)}$/- |
| + reasoning | 78.34$_{(5)}$/- | 89.80$_{(15)}$/- | 66.00$_{(10)}$/- | 73.20$_{(13)}$/- |
| *CoT-ER* (0% training data) | | | | |
| Auto-CoT-ER | 85.40$_{(5)}$/- | 93.40$_{(13)}$/- | 76.10$_{(10)}$/- | 86.4$_{(13)}$/- |
| Manual-CoT-ER | 88.00$_{(5)}$/- | - | 82.60$_{(10)}$/- | - |

Table 2: Main Results on FewRel 2.0 validation/test set. All results are given by accuracy (%). $(m)$ means there are $m$ instances selected as few-shot demonstrations. $+reasoning$ denotes the demand of reasoning process generation.

few-shot scenario.

Third, CoT-ER prompting method achieves competitive performance compared to the state-of-the-art fully-supervised method and surpasses the majority of fully-supervised methods with minimal manual labor both on FewRel 1.0 and FewRel 2.0. This suggests that GPT series LLMs have the potential to beat previous fully-supervised methods when high-quality relation information and well-designed reasoning processes are provided.

## 5.2 Ablation Study on CoT-ER

Does the incorporation of entity information significantly benefit the coT-ER? To this end, we conducted ablation experiments to demonstrate the necessity of the 3-step reasoning process. In this experiment, we removed the first and second steps and compared the performance with Auto-CoT-reasoning. For fairness concerns, we implemented this experiment using Auto-CoT-ER, which also employs an auto-generated reasoning process by LLM. Due to the limitation of maximum input and output tokens, we set the number of instances in the few-shot demonstrations to 13 for the ablation experiments. The results are presented in Figure 4.

We find that: (1) after removing the first and second steps, the performance of Auto-CoT-ER shows a significant decline with reductions of 3.4, 2.2, 1.8, 2.9, and 5.2, 6, 5.3, 7.6 Accuracy on FewRel 1.0 and FewRel 2.0 respectively. It means higher-level

abstraction of entity types, specifically the concept-level entities, are beneficial to the LLM performing RE task in the few-shot scenario. (2) Despite the third step of CoT-ER pairing the support instance with a simpler reasoning process compared to Auto-CoT, it achieves superior performance in certain challenging scenarios (10-Way 1-Shot and medical domain §5.1). This finding indicates that the semantic information provided by the relation label is more beneficial to the LLM than low-quality reasoning information.

## 5.3 The Stability of CoT-ER

**Different Random Seeds for Task Sampling.** Due to the high cost of the "text-davinci-003", we sample a relatively small number of queries for testing, specifically $100 \times N$ for each N-Way K-Shot task. It may raise concerns that the results may not hold up when evaluated on the full test sets. To this end, we evaluated the CoT-ER and Vanilla-ICL using 8 random seeds for N-Way K-Shot task sampling. Table 3 and Table 4 show experimental results with mean ± standard deviation on FewRel2.0. Notably, CoT-ER consistently outperforms Vanilla-ICL across all N-way K-shot settings with a lower standard deviation.

**Different Number of Few-shot Instances.** To investigate how the selected number of demonstrations contribute to the performance of CoT-ER,

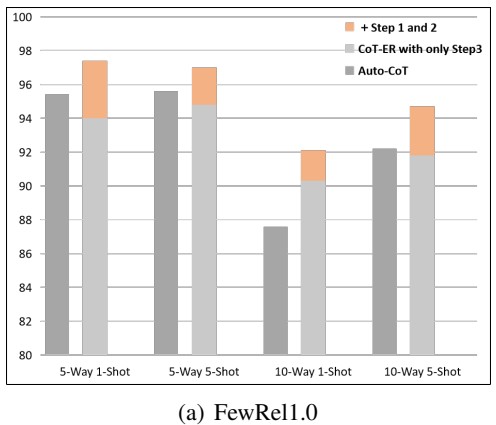

(a) FewRel1.0

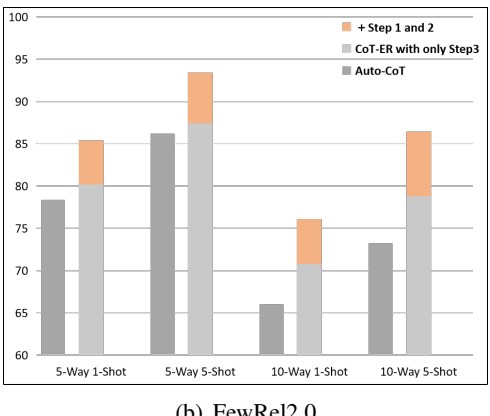

(b) FewRel2.0

Figure 4: Ablation study on the first and second reasoning steps of CoT-ER. Auto-CoT refers to the "with reasoning generation" version.

| Method | 5-Way 1-Shot | 5-Way 5-Shot |
|---|---|---|
| **Vanilla-ICL** | 80.75±1.70 | 88.19±1.56 |
| **CoT-ER** | 85.64±1.28 | 92.99±1.43 |

Table 3: 5-way classification performance comparison (mean ± standard deviation) between CoT-ER and Vanilla-ICL on FewRel 2.0 across 8 different random seeds.

| Method | 10-Way 1-Shot | 10-Way 5-Shot |
|---|---|---|
| **Vanilla-ICL** | 68.21±1.27 | 80.92±1.05 |
| **CoT-ER** | 77.61±0.87 | 86.18±0.88 |

Table 4: 10-way classification performance comparison (mean ± standard deviation) between CoT-ER and Vanilla-ICL on FewRel 2.0 across eight different random seeds.

we conducted experiments across different $\mathcal{M}$ under the 5-Way 5-Shot setting. A single prompt can hold 13 CoT-ER reasoning demonstrations at worst, whereas all the support instances (25) can be appended to the prompt in Vanilla-ICL. The results are presented in Table 5.

We observe that both CoT-ER and Vanilla-ICL can benefit from more few-shot examples, which aligns with the same conclusion in previous work (Liu et al., 2022a). However, the performance of Vanilla-ICL deteriorates rapidly as the number of examples decreases. CoT-ER can effectively leverage the information from provided instances and maintain strong performance even with a reduced number of instances. This demonstrates that CoT-ER exhibits greater stability compared to Vanilla-ICL when the number of few-shot instances varies.

## 5.4 Case Study

We select one typical reasoning example generated by LLM to better demonstrate the superiority of our prompting method CoT-ER. As shown in Table 6, this instance necessitates the LLM to correctly identify the relation label "crosses" between "Railway Bridge" and "Daugava". In FewRel 1.0, the relation label "crosses" is described as *"obstacle (body of water, road, ...) which this bridge crosses over or this tunnel goes under"*.

However, using Auto-CoT prompting leads to a wrong prediction where the model incorrectly labels the relation with "located in or next to body of water", which pertains to the relationship between a location entity and a water-related entity (river, lake, ...). The primary reason for the failure of Auto-CoT is the absence of higher-level abstraction of entities in reasoning, which is necessary to comprehend the entities involved in the relation. CoT-ER addresses this issue by incorporating concept-level information into the reasoning process through its first and second steps. Specifically, in this case, the LLM first reasons the subject entity corresponds to a bridge and the object entity corresponds to a river based on its own knowledge base and contextual information, thereby excluding the relation requiring a location entity and water-related entities (as demonstrated by other examples in prompt). With this clue, the LLM can effectively perform subsequent reasoning steps.

Furthermore, the presence of both "crosses" and "located in or next to body of water" labels in an N-way K-shot task can indeed confuse the LLM due to the lack of semantic information on these two phrases. CoT-ER addresses this issue by inte-

| $\mathcal{M}$ | 5 | 6 | 7 | 8 | 9 | 10 | 11 | 12 | 13 | 15 | 17 | 19 | 21 | 23 | 25 |
|---|---|---|---|---|---|---|---|---|---|---|---|---|---|---|---|
| **CoT-ER** | 88.40 | 88.80 | 90.80 | 91.60 | 91.20 | 89.20 | 89.20 | 91.80 | 94.20 | - | - | - | - | - | - |
| **Vanilla-ICL** | 73.40 | 78.00 | 78.80 | 82.90 | 81.40 | 82.39 | 81.20 | 82.79 | 84.20 | 87.60 | 89.0 | 90.40 | 90.80 | 90.60 | 89.80 |

Table 5: Performance comparison (given by accuracy) between CoT-ER and Vanilla-ICL on FewRel 2.0 across different numbers of demonstrations in a single prompt.

| Prompting Method | Reasoning Process | Correct |
|---|---|---|
| Auto-CoT | The clues that lead to the relation between "Railway Bridge" and "Daugava" to be "located in or next to body of water" in the sentence are the words "crosses" and "river". These words indicate that the Railway Bridge is crossing the Daugava River, which is a body of water. | ✗ |
| CoT-ER | 1. Subject entity "Railway Bridge" is a bridge name, which refers to the entity of bridge in the context. 
 2. Object entity "Daugava" is a river name, which refers to the entity of river in the context. 
 3. According to the context, "The Railway Bridge is a bridge that crosses the Daugava river." indicates that "Railway Bridge" crosses "Daugava". | ✓ |

Table 6: We compare our prompting method CoT-ER with Auto-CoT for an example in FewRel 1.0. The context of this example is "*The Railway Bridge is a bridge that crosses the Daugava river in Riga, the capital of Latvia.*", and the relation between two highlight entities is "crosses". The key reasoning processes of CoT-ER are highlighted with green.

grating both entities and the relation label into a coherent expression, as exemplified by this example *"Railway Bridge" crosses "Daugava"*.

# 6 Conclusion

In this paper, we explore the potential of LLM in-context learning on few-shot relation extraction. To enhance the general performance caused by low-quality auto-generated reasoning processes, we introduce CoT-ER, a prompting method tailored to few-shot relation extraction. The core idea is to prompt LLM to generate evidence using task-specific and concept-level knowledge stored in its pre-training stage. These pieces of evidence will be utilized by LLM during the RE task, and facilitate the reasoning process. Additionally, we devise a label verbalizing technique by integrating both entities and the relation label into a coherent expression. This technique addresses the semantic ambiguity of relation labels, which is a common challenge encountered during relation extraction when utilizing in-context learning. The experimental results on FewRel 1.0 and FewRel 2.0 outperform all training-free baselines, demonstrating the effectiveness of our proposed approach. Moreover, achieving comparable results to the state-of-the-art fully-supervised method suggests that the paradigm of in-context learning holds promise as a novel solution for the few-shot relation extraction task.

# Limitations

Although CoT-ER achieved decent results on FewRel 1.0 and FewRel 2.0, there is still poten-tial for future improvement. Our proposed method does not fully utilize all instances when handling larger support sets, such as 5-way 5-shot and 10-way 5-shot, due to the constraint of maximum length. Though we adopt a similarity-based KNN retrieval to select superior instances for few-shot demonstrations, we find it not as effective in the few-shot setting compared to other works that perform well when there is a large candidate set available. Due to the high cost of employing reasoning-required ICL via GPT-3 API, we have not evaluated the CoT-ER on a superior LLM with longer maximum input tokens and a larger scale.

Our limited budget also restricted the optimization for the construction of seed examples. It is possible to enhance the performance with a more informative and appropriate design.

# Ethics Statement

It is known that pre-trained language models could capture the bias reflecting training data. Thus, our approach using LLMs can potentially generate offensive or biased content. We acknowledge this risk and suggest that practitioners should carefully examine the potential bias before deploying our models in real-world applications.

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

# A  Error Analysis

## A.1  Error Analysis of Relation Type

To fully justify CoT-ER as a superior approach to the baseline, we will provide a quantitative analysis comparing CoT-ER and Vanilla-ICL across all test classes. Taking the 5-Way 1-Shot setting as an example, Table 7 and Table 8 display the accuracy of these two methods across various relation classes within FewRel. Note that different numbers represent different relation types, and their order is based on the performance ranking of Vanilla-ICL. These accuracies are the average results obtained from multiple test runs, each performed with seven different random seeds.

Table 7 shows the experimental results on FewRel 1.0. We can observe that CoT-ER surpasses

| Relation Type | 1 | 2 | 3 | 4 | 5 | 6 | 7 | 8 | 9 | 10 | 11 | 12 | 13 | 14 | 15 | 16 |
|---|---|---|---|---|---|---|---|---|---|---|---|---|---|---|---|---|
| Vanilla-ICL | 100 | 100 | 99.12 | 98.63 | 98.3 | 98.13 | 97.72 | 96.64 | 96.57 | 94.15 | 92.94 | 91.01 | 85.24 | 84.63 | 83.71 | 82.47 |
| CoT-ER | 100 | 100 | 99.52 | 98.07 | 100 | 98.65 | 95.51 | 89.47 | 100 | 95.25 | 91.72 | 95.66 | 89.31 | 95.98 | 86.95 | 91.39 |
| Difference | 0 | 0 | **0.4** | -0.56 | **1.7** | **0.52** | -2.21 | -7.17 | **3.43** | **1.1** | -1.22 | **4.65** | **4.07** | **11.35** | **3.24** | **8.92** |

Table 7: Results of each label in FewRel 1.0. Definition of digital relation type:1 voice type, 2 position played on team/speciality, 3 original language of film or TV show, 4 constellation, 5 military rank, 6 competition class, 7 member of, 8 spouse, 9 located in or next to body of water, 10 follows, 11 crosses, 12 sport, 13 main subject, 14 child, 15 part of, 16 mother.

| Relation Type | 1 | 2 | 3 | 4 | 5 | 6 | 7 | 8 | 9 | 10 |
|---|---|---|---|---|---|---|---|---|---|---|
| Vanilla-ICL | 98.71 | 95.71 | 88.82 | 88.19 | 79.33 | 75.54 | 74.07 | 70.71 | 67.69 | 65.46 |
| CoT-ER | 97.84 | 95.95 | 89.29 | 87.3 | 71.32 | 93.01 | 71.32 | 89.84 | 80.41 | 81.16 |
| Difference | -0.87 | **0.24** | **0.47** | -0.89 | -8.01 | **17.47** | -2.75 | **19.13** | **12.72** | **15.7** |

Table 8: Results of each label in FewRel 2.0. Definition of digital relation type: 1 inheritance type of, 2 ingredient of, 3 classified as, 4 gene found in organism, 5 is primary anatomic site of disease, 6 causative agent of, 7 biological process involves gene product, 8 gene plays role in process, 9 is normal tissue origin of disease, 10 occurs in.

| Relation label | Reasoning Process | Correct |
|---|---|---|
| gene plays role in process | Context:
In the hippocampus, transcriptional upregulation was observed in nogo-a(one day post-injury), mag and pirb at seven days post-injury.
Reasoning process of CoT-ER:
1. **Entity "transcriptional" refers to the process of transcribing DNA into RNA**, which refers to the entity of biological process in the context.
2. **Entity "mag" is the name of a gene**, which refers to the entity of gene in the context.
3. According to the context, "transcriptional upregulation was observed in nogo-a (one day post-injury), mag and pirb at seven days post-injury" indicates that **"mag" plays role in "transcriptional"**.
So, the relation between "transcriptional" and "mag" is "gene plays role in process". | ✓ |
| spouse | Context:
Their maternal grandparents were John II, Count of Holland and Philippa of Luxembourg.
Reasoning process of CoT-ER:
1. **Subject entity "Philippa of Luxembourg" is a personal name**, which refers to the entity of person in the context.
2. **Object entity "John II, Count of Holland" is a personal name**, which refers to the entity of person in the context.
3. According to the context, "maternal grandparents" indicates that Object entity "John II, Count of Holland" is a child of Subject entity "Philippa of Luxembourg".
So, the relation between "Philippa of Luxembourg" and "John II, Count of Holland" is "child". | ✗ |
| is primary anatomic site of disease | Context:
Loss of hair in the other areas of the skin is present in the majority of cases.
Reasoning process of CoT-ER:
1. **Entity "Loss of hair" is the name of a condition**, which refers to the entity of condition in the context.
2. **Entity "skin" is the name of a body organ**, which refers to the entity of physical location of body part in the context.
3. According to the context, "loss of hair in the other areas of the skin" indicates that "loss of hair" occurs in "skin".
So, the relation between "loss of hair" and "skin" is "occurs in". | ✗ |

Table 9: The representative scenarios where CoT-ER makes mistakes for different relation class.

Vanilla-ICL 10 times, with 7 of them showing a relatively high improvement. However, in the 7th and 8th relations, CoT-ER still lags behind Vanilla-ICL by a few percentage points.

Table 8 shows the experimental results on FewRel 2.0. We can observe that CoT-ER surpasses Vanilla-ICL 6 times, with 4 of them showing a significant improvement of over 10%. However, in the 5th and 7th relations, CoT-ER still lags behind Vanilla-ICL by a few percentage points.

## A.2 Error Analysis of Reasoning Process

CoT-ER shows significant improvements in scenarios when Vanilla-ICL performs poorly. Here, we present a few cases to illustrate the reasoning process of CoT-ER in some representative scenarios in order to facilitate future research. Table 9 shows some correct and incorrect answers produced by CoT-ER.

Relation class **gene plays role in process**: In this case, CoT-ER can not only recognize what "transcriptional" and "mean" mean in the context but also have concept-level knowledge. And the final prediction is correct.

Relation class **spouse**: In this case, CoT-ER correctly recognizes the entity type and precisely extracts the crucial evidence "maternal grandparents". However, the LLM incorrectly interprets "maternal grandparents" as the relationship between these two entities, when they are actually a couple of "maternal grandparents". This demonstrates that the LLM with CoT-ER may overlook contextual information sometimes.

Relation class **is primary anatomic site of disease**: In this case, CoT-ER can also recognize the entity well, and the conclusion is semantically right

("loss of hair" occurs in "skin"). However, the final prediction is incorrect, as the predicted relation label is "occurs in" while the ground truth label is "is primary anatomic site of disease". The reason is that the label "occurs in" in this dataset means "a condition occurs in a period of the lifetime". This particular label should be matched with entity pairs like "condition or disease" and "a period of a person's lifetime (such as congenital)". This issue indicates that the relation description would be a key component in such methods, but it's not included in the FewRel 2.0 dataset.

## B  Seed Examples

In this section, we will give more details about how we construct the seed examples and present all seed examples in Table 10, Table 11, Table 12, Table 13 and Table 14.

We first randomly select one instance from each relation class to serve as the seed example. Then we outline the three steps of CoT-ER in each instance. By considering the contextual information of the two entities involved, we manually assign the entity types and identify relevant text spans as evidence. Because FewRel 1.0 provides a description of each relation (such as "sport: sport in which the subject participates or belongs to") in a separate file while FewRel 2.0 does not, this has implications for the design of seed examples. However, these descriptions are not directly used in the prompts. Furthermore, we haven't optimized the seed examples, leaving room for further improvement.

## C  Prompts

All prompt templates used in this paper are presented in Table 15.

**mother**
Context:
Anne de Bourbon (1380–September 1408) was a daughter of John I, Count of La Marche and his wife Catherine of Vendôme.

CoT-ER:
1. Subject entity "Anne de Bourbon" is a personal name, which refers to the entity of son or daughter in the context.
2. Object entity "Catherine of Vendôme" is a personal name, which refers to the entity of female human beings in the context.
3. According to the context, "was a daughter of John I, Count of La Marche" and "his wife" indicate that
   object entity "Catherine of Vendôme" is subject entity "Anne de Bourbon"'s mother.
So, the relation between subject entity "Anne de Bourbon" and object entity "Catherine of Vendôme" is "mother".

**child**
Context:
He was a son of Margrethe Rode, and a brother of writer Helge Rode.

CoT-ER:
1. Subject entity "Margrethe Rode" is a personal name, which refers to the entity of an adult in the context.
2. Object entity "Helge Rode" is a personal name, which refers to the entity of son or daughter in the context.
3. According to the context, "son of Margrethe Rode" and "brother of writer Helge Rode" indicate that
   Object entity "Helge Rode" is a child of Subject entity "Margrethe Rode".
So, the relation between subject entity "Margrethe Rode" and object entity "Helge Rode" is "child".

**spouse**
Context:
Hanke's seemingly unstoppable ascent on the coattails of Goebbels came to a sudden, albeit temporary,
halt when he was drawn into the marital affairs of Joseph Goebbels and his wife, Magda.

CoT-ER:
1. Subject entity "Magda" is a personal name, which refers to the entity of human beings in the context.
2. Object entity "Joseph Goebbels" is a personal name, which refers to the entity of human beings in the context.
3. According to the context, "marital affairs" and "his wife" indicate that
   "Magda" is "Joseph Goebbels"'s wife, which means they are a married couple.
So, the relation between subject entity "Magda" and object entity "Joseph Goebbels" is "spouse".

**sport**
Context:
Demetrius Rhaney (born June 22, 1992) is an American football center for the Washington Redskins of
the National Football League (NFL).

CoT-ER:
1. Subject entity "Demetrius Rhaney" is a personal name, which refers to the entity of American football player in the context.
2. Object entity "American football" is a sport, which refers to the entity of sport in the context.
3. According to the context, "Demetrius Rhaney (born June 22, 1992) is an American football center" indicates that
   "Demetrius Rhaney" participates or belongs to "American football".
So, the relation between subject entity "Demetrius Rhaney" and object entity "American football" is "sport".

**crosses**
Context:
Wilton Bridge was a major crossing of the River Wye and was protected by Wilton Castle.

CoT-ER:
1. Subject entity "Wilton Bridge" is a bridge name, which refers to the entity of bridge in the context.
2. Object entity "River Wye" is a river name, which refers to the entity of river in the context.
3. According to the context, "Wilton Bridge was a major crossing of the River Wye" indicates that
   "Wilton Bridge" crosses "River Wye".
So, the relation between subject entity "Wilton Bridge" and object entity "River Wye" is "crosses".

**original language of film or TV show**
Context:
He composed a song called "Zindagi aur bata tera irada kya hai" for the Hindi movie "Zindagi Aur Toofan" (1975).

CoT-ER:
1. Subject entity "Zindagi Aur Toofan" is a movie name, which refers to the entity of a film or a performance work
   in the context.
2. Object entity "Hindi" is a language, which refers to the entity of language in the context.
3. According to the context, "Hindi movie Zindagi Aur Toofan" indicates that
   the film "Zindagi Aur Toofan" was originally made in "Hindi".
So, the relation between subject entity "Zindagi Aur Toofan" and object entity "Hindi" is
   "original language of film or TV show".

Table 10: Seed examples in FewRel 1.0.

**competition class**
Context:
Sasakul turned pro in 1991 and captured the WBC and lineal flyweight titles with a win over Yuri Arbachakov in 1997.

CoT-ER:
1. Subject entity "Yuri Arbachakov" is a personal name, which refers to the entity of events, teams, participants, or equipment in the context.
2. Object entity "flyweight" is a competition type of boxing match, which refers to the entity of competition or organization in the context.
3. According to the context, "captured the WBC and lineal flyweight titles with a win over Yuri Arbachakov" indicates that object entity "Yuri Arbachakov" qualifies for inclusion in "flyweight".
So, the relation between subject entity "Yuri Arbachakov" and object entity "flyweight" is "competition class".

---

**part of**
Context:
The observatory began operation around 1984 with the Isaac Newton Telescope, which was moved to La Palma from the Royal Greenwich Observatory site at Herstmonceux Castle in Sussex, England.

CoT-ER:
1. Subject entity "Isaac Newton Telescope" is a scientific instrument, which refers to the entity of components in the context.
2. Object entity "Royal Greenwich Observatory" is scientific institution, which refers to the entity of entirety in the context.
3. According to the context, "which was moved to La Palma from the Royal Greenwich Observatory" indicates that "Isaac Newton Telescope" is part of "Royal Greenwich Observatory".
So, the relation between subject entity "Isaac Newton Telescope" and object entity "Royal Greenwich Observatory" is "part of".

---

**constellation**
Context:
Tau2 Gruis, is a double star located in the constellation Grus.

CoT-ER:
1. Subject entity "Tau2 Gruis" is the name of a celestial body, which refers to the entity of a celestial body in the context.
2. Object entity "Grus" is the name of a constellation, which refers to the entity of the constellation in the context.
3. According to the context, "located in the constellation Grus" indicates that "Tau2 Gruis" is part of constellation "Grus".
So, the relation between subject entity "Tau2 Gruis" and object entity "Grus" is "constellation".

---

**position played on team/speciality**
Context:
Haitham Simreen(born 1 January 1977) is a retired Jordanian footballer of Palestinian origin, who was a Defender for Al-Wehdat and the Jordan national football team.

CoT-ER:
1. Subject entity "Haitham Simreen" is a personal name, which refers to the entity of sports player in the context.
2. Object entity "Defender" is a position in football, which refers to the entity of position or specialism in the context.
3. According to the context, "who was a Defender for Al-Wehdat and the Jordan national football team" indicates that "Haitham Simreen" plays as a "Defender" position in that team.
So, the relation between subject entity "Haitham Simreen" and object entity "defender" is "position played on team/speciality".

---

**located in or next to body of water**
Context:
Both the Grand Canal, and the Royal Canal flow through Westmeath, and the River Shannon (Ireland's key tourism waterway) has a modern inland harbour in Athlone.

CoT-ER:
1. Subject entity "Athlone" is a place name, which refers to the entity of position in the context.
2. Object entity "River Shannon" is the name of the river, which refers to the entity of river or water in the context.
3. According to the context, "River Shannon (Ireland's key tourism waterway) has a modern inland harbour in Athlone" indicates that "Athlone" is located in or next to body of water "River Shannon".
So, the relation between subject entity "Athlone" and object entity "River Shannon" is "located in or next to body of water".

---

**voice type**
Context:
Gabriella Gatti (July 5, 1908 - October 22, 2003) was an Italian operatic soprano, primarily based in Italy and associated with the Italian repertory.

CoT-ER:
1. Subject entity "Gabriella Gatti" is a personal name, which refers to the entity of a person in the context.
2. Object entity "soprano" is a tone, which refers to the entity of voice type in the context.
3. According to the context, "Italian operatic soprano" indicates that "Gabriella Gatti" has a voice type of "soprano".
So, the relation between subject entity "Gabriella Gatti" and object entity "soprano" is "voice type".

Table 11: Seed examples in FewRel 1.0.

**follows**
Context:
Hot Dance Club Play chart, along with album tracks "Whammy Kiss" and "Song for a Future Generation".

CoT-ER:
1. Subject entity "Song for a Future Generation" is a mention in the context, which refers to the entity of part of a series in the context.
2. Object entity "Whammy Kiss" is a mention in the context, which refers to the entity of part of series in the context.
3. According to the context, "Whammy Kiss and Song for a Future Generation" indicates that
   "Song for a Future Generation" follows "Whammy Kiss".
So, the relation between subject entity "Song for a Future Generation" and object entity "Whammy Kiss" is "follows".

**military rank**
Context:
The French under Jean de Vienne, Admiral of France joined forces with the Scots.

CoT-ER:
1. Subject entity "Jean de Vienne" is a personal name, which refers to the entity of a person in the context.
2. Object entity "Admiral of France" is a name of military rank, which refers to the entity of military rank in the context.
3. According to the context, "Jean de Vienne, Admiral of France" indicates that
   "Jean de Vienne" holds the military rank of "Admiral of France".
So, the relation between subject entity "Jean de Vienne" and object entity "Admiral of France" is "military rank".

**member of**
Context:
Neil Tennant and Chris Lowe of the Pet Shop Boys commented on this remix to journalist Mark Beaumont,
writing for the "NME", in February 2017.

CoT-ER:
1. Subject entity "Chris Lowe" is a personal name, which refers to the entity of a person in the context.
2. Object entity "Pet Shop Boys" is the name of the organization, which refers to the entity of the organization or club in the context.
3. According to the context, "Chris Lowe of the Pet Shop Boys" indicates that
   "Chris Lowe" is a member of "Pet Shop Boys"
So, the relation between subject entity "Chris Lowe" and object entity "Pet Shop Boys" is "member of".

**main subject**
Context:
The British Free Corps is featured in Jack Higgins' World War II thriller "The Eagle Has Landed".

CoT-ER:
1. Subject entity "The Eagle Has Landed" is an artistic work name, which refers to the entity of work in the context.
2. Object entity "World War II" is a historical event, which refers to the entity of the main subject or theme in the context.
3. According to the context, "World War II thriller "The Eagle Has Landed"" indicates that
   "World War II" is the main subject of the work named "The Eagle Has Landed".
So, the relation between subject entity "The Eagle Has Landed" and object entity "World War II" is "main subject".

Table 12: Seed examples in FewRel 1.0.

**biological process involves gene product**
Context:
I have discussed various aspects of the structure, regulation and function of the protein tyrosine phosphatase family,
which I hope will illustrate the fundamental importance of these enzymes in the control of signal transduction.

CoT-ER:
1. Entity "protein tyrosine phosphatase" is a type of enzyme, which refers to the entity of gene product in the context.
2. Entity "signal transduction" is the name of biological process, which refers to the entity of biological process in the context.
3. According to the context, "the fundamental importance of these enzymes in the control of signal transduction" indicates that
   "protein tyrosine phosphatase" is involved in "signal transduction".
So, the relation between "protein tyrosine phosphatase" and "signal transduction" is
"biological process involves gene product".

---

**gene plays role in process**
Context:
We demonstrate here that the critical step in establishing the angiogenic capability of human tumor cells is
the repression of a key secreted anti-angiogenic factor, thrombospondin-1 (tsp-1).

CoT-ER:
1. Entity "angiogenic" is the name of a biological process, which refers to the entity of biological process in the context.
2. Entity "tsp-1" is an abbreviation of thrombospondin-1, which refers to the entity of gene product in the context.
3. According to the context, "establishing the angiogenic capability" and "the repression of a key secreted
   anti-angiogenic factor, thrombospondin-1" indicate that "tsp-1" plays role in establishing the "angiogenic" capability.
So, the relation between "angiogenic" and "tsp-1" is "gene plays role in process".

---

**occurs in**
Context:
We determined whether this is also true for the congenital muscle disease nemaline myopathy using our mouse model of
this disease.

CoT-ER:
1. Entity "congenital" refers to a condition or trait that is present at or before birth, which refers to the entity of time interval
   in the context.
2. Entity "nemaline myopathy" is the name of a disease, which refers to the entity of disease in the context.
3. According to the context, "congenital muscle disease nemaline myopathy" indicates that
   "nemaline myopathy" occurs in "congenital" period.
So, the relation between "congenital" and "nemaline myopathy" is "occurs in".

---

**inheritance type of**
Context:
Here we are presenting two siblings, a 14-year-old male and a 13-year-old female with features of progeria,
suggesting a possible autosomal recessive inheritance.

CoT-ER:
1. Entity "progeria" is the name of a disease, which refers to the entity of the disease in the context.
2. Entity "autosomal recessive inheritance" is the name of a specific pattern of inheritance, which refers to
   the entity of inheritance in the context.
3. According to the context, "suggesting a possible autosomal recessive inheritance" indicates that
   "progeria" is a type of "autosomal recessive inheritance" disease.
So, the relation between "progeria" and "autosomal recessive inheritance" is "inheritance type of".

---

**is normal tissue origin of disease**
Context:
microcystic adnexal carcinoma (mac) is known as an infiltrating but non-metastasizing tumour of the skin,
that derives from sweat glands or follicular epithelium.

CoT-ER:
1. Entity "microcystic adnexal carcinoma" is the name of a disease, which refers to the entity of disease in the context.
2. Entity "epithelium" is a type of cell, which refers to the entity of biological tissue in the context.
3. According to the context, "that derives from sweat glands or follicular epithelium" indicates that
   "epithelium" is the normal tissue origin of "microcystic adnexal carcinoma".
So, the relation between "microcystic adnexal carcinoma" and "epithelium" is "is normal tissue origin of disease".

---

**causative agent of**
Context:
The pathogenesis of amoebic dysentery is a result of cytolysis of the colonic mucosa by the parasitic protozoan
entamoeba histolytica.

CoT-ER:
1. Entity "amoebic dysentery" is the name of a disease, which refers to the entity of disease in the context.
2. Entity "entamoeba histolytica" is a type of protozoa, which refers to the entity of causative agent in the context.
3. According to the context, "is a result of cytolysis of the colonic mucosa by the parasitic protozoan entamoeba histolytica"
   indicates that "entamoeba histolytica" is the causative agent of "amoebic dysentery".
So, the relation between "amoebic dysentery" and "entamoeba histolytica" is "causative agent of".

Table 13: Seed examples in FewRel 2.0.

**classified as**

Context:

On the other hand, in the nonlethal p. yoelii 17xnl infection, wt mice could control a primary infection with $1 \times 10$ (5) parasitized erythrocytes.

CoT-ER:

1. Entity "infection" is the name of a biological process, which refers to the entity of abstractions of a certain type of thing in the context.
2. Entity "primary infection" is the name of a biological process, which refers to the entity of abstractions of a certain type of thing in the context.
3. According to the context, "in the nonlethal p. yoelii 17xnl infection" and "control a primary infection with $1 \times 10$ (5) parasitized erythrocytes" indicate that "infection" and "primary infection" are related to each other in a way that one is a subset of the other or two classes of something.

So, the relation between "infection" and "primary infection" is "classified as".

**gene found in organism**

Context:

Ustekinumab is a fully human monoclonal immunoglobulin antibody that targets the interleukin (il) -12 and il-23 shared p40 subunit.

CoT-ER:

1. Entity "human" is a type of organism, which refers to the entity of an organism in the context.
2. Entity "il-23" is the name of a gene, which refers to the entity of the gene in the context.
3. According to the context, "ustekinumab is a fully human monoclonal immunoglobulin antibody" and "targets the interleukin (il) -12 and il-23 shared p40 subunit" indicate that "il-23" is found in "human" organism.

So, the relation between "human" and "il-23" is "gene found in organism".

**ingredient of**

Context:

Against a bla+ isolate, the combination of piperacillin with tazobactam with streptomycin resulted in a synergistic effect relative to that of piperacillin with tazobactam ; piperacillin plus streptomycin did not show synergism.

CoT-ER:

1. Entity "piperacillin with tazobactam" is a combination of two entities, which refers to the entity of composite component in the context.
2. Entity "piperacillin" is the name of a drug, which refers to the entity of individual components in the context.
3. According to the context, "the combination of piperacillin with tazobactam" indicates that "piperacillin" is ingredient of "piperacillin with tazobactam".

So, the relation between "piperacillin with tazobactam" and "piperacillin" is "ingredient of".

**is primary anatomic site of disease**

Context:

Twenty-five of 88 (28.4%) had endometrial carcinoma on final uterine pathology.

CoT-ER:

1. Entity "endometrial carcinoma" is the name of a disease, which refers to a medical condition in the context.
2. Entity "uterine" is the name of the body organ, which refers to the entity of physical location of body parts in the context.
3. According to the context, "endometrial carcinoma on final uterine pathology" indicates that "uterine" is primary anatomic site of "endometrial carcinoma".

So, the relation between "endometrial carcinoma" and "uterine" is "is primary anatomic site of disease".

Table 14: Seed examples in FewRel 2.0.

**Vanilla-ICL**
Please solve the Relation Extraction task.
Given the context, consider what's the most precise relation between two entities belonging to the following [N] possible relations.
The relation must be in these [N] possible relations: relation label 1, ..., relation label N

Context: [CONTEXT]
Given the context, the relation between [HEAD_ENTITY] and [TAIL_ENTITY] is [GOLDEN_LABEL].
$\times \mathcal{M}$

Context: [CONTEXT]
Given the context, the relation between [HEAD_ENTITY] and [TAIL_ENTITY] is

**Auto-CoT**
Please solve the Relation Extraction task.
Given the context, consider what's the most precise relation between two entities belonging to the following [N] possible relations.
The relation must be in these [N] possible relations: relation label 1, ..., relation label N

Context: [CONTEXT]
Given the context, what's the relation between [HEAD_ENTITY] and [TAIL_ENTITY]?
[Answer with CoT]
So the relation between [HEAD_ENTITY] and [TAIL_ENTITY] is [GOLDEN_LABEL].
$\times \mathcal{M}$

Context: [CONTEXT]
Given the context, what's the relation between [HEAD_ENTITY] and [TAIL_ENTITY]?

**CoT-generation**
Please solve the Relation Extraction task.
Given the context, figure out the reasoning steps that lead to the relation between two entities to be the specific one.

Context: [CONTEXT]
Given the context, what's the relation between [HEAD_ENTITY] and [TAIL_ENTITY]?
Now, known the relation is [GOLDEN_LABEL], the reasoning steps are:
[Seed example of the specific class]
So the relation between [HEAD_ENTITY] and [TAIL_ENTITY] is [GOLDEN_LABEL].
$\times \mathcal{M}$

Context: [CONTEXT]
Given the context, what's the relation between [HEAD_ENTITY] and [TAIL_ENTITY]?
Now, known the relation is [GOLDEN_LABEL], the reasoning steps are:

Table 15: Prompt templates in the paper.