# OpenReview forum: "Chain of Thought with Explicit Evidence Reasoning for Few-shot Relation Extraction"
_EMNLP/2023/Conference — EMNLP 2023 Findings_

### Official Review · Reviewer_6ESd · 2023-08-01

**Soundness:** 3

**Excitement:**

3: Ambivalent: It has merits (e.g., it reports state-of-the-art results, the idea is nice), but there are key weaknesses (e.g., it describes incremental work), and it can significantly benefit from another round of revision. However, I won't object to accepting it if my co-reviewers champion it.

**Paper Topic And Main Contributions:**

This paper proposes a novel approach for few-shot relation extraction called CoT-ER, which utilizes large language models and explicit evidence reasoning to improve performance. The proposed method achieves competitive performance compared to fully-supervised approaches, without the need for training data. Experimental results demonstrate the effectiveness of the proposed approach on the FewRel1.0 and FewRel2.0 datasets.

**Reasons To Accept:**

1.	The paper proposes a novel CoT-ER method to incorporate task-specific and concept-level knowledge into chain-of-thought prompting for relation extraction.

2.	The paper is well-organized and clearly presents the proposed framework.

3.	The experimental results are promising and show significant improvement over the compared models.


**Reasons To Reject:**

1.	The paper does not provide a clear explanation of how the prompts used in CoT-ER are selected. The authors should provide more details on how different prompts were evaluated and how they affected the experimental results. It would be better to give a more thorough analysis of the impact of different prompts on the performance of the proposed method.
2.	Given that large language models have the potential to achieve high performance on relation extraction tasks, it would be valuable for the authors to conduct a thorough error analysis to identify the reasons for any mistakes made by the model. This analysis could provide insights into the limitations of the proposed approach and suggest directions for future research.
3.	The paper does not provide a detailed analysis of the impact of the k value on the performance of the proposed method. k-NN search is a key component, a more thorough analysis of the impact of the k value on the experimental results is needed.


**Reproducibility:**

4: Could mostly reproduce the results, but there may be some variation because of sample variance or minor variations in their interpretation of the protocol or method.

**Reviewer Confidence:**

4: Quite sure. I tried to check the important points carefully. It's unlikely, though conceivable, that I missed something that should affect my ratings.

---

> ### Author Rebuttal · Authors · 2023-08-29
>
> Thank you for your valuable comments. We will explain your concerns point by point.
>
> 1. **Comment**: The paper does not provide a clear explanation of how the prompts used in CoT-ER are selected. The authors should provide more details on how different prompts were evaluated and how they affected the experimental results. It would be better to give a more thorough analysis of the impact of different prompts on the performance of the proposed method.
>
>    The paper does not provide a detailed analysis of the impact of the k value on the performance of the proposed method. k-NN search is a key component, a more thorough analysis of the impact of the k value on the experimental results is needed.
>
>    **Reply**: There are two perspectives to consider different prompts. **One is the wording, another is the in-context examples (the k value)**. For the wording of the prompt, we adhere to the principle of **extreme simplicity** in crafting prompts for all methods in the paper. It means we do not introduce additional sentences with task-irrelevant descriptions, aiming to avoid prompt engineering.
>
>    For the selection of in-context examples (the k value), we evaluate **the effect of the selected number** of instances in prompt and have conducted an ablation experiment. We show the scenario of 5-Way 5-Shot on FewRel2.0 here, where the CoT-ER method can hold up to 13 instances in prompt while Naive-ICL can hold all the support set (25) instances in prompt. More details will be added to the revised paper.
>
>    | #instances of prompt (k value) | 5     | 6     | 7     | 8     | 9     | 10    | 11    | 12    | 13    |
>    | ------------------------------ | ----- | ----- | ----- | ----- | ----- | ----- | ----- | ----- | ----- |
>    | CoT-ER                         | 88.40 | 88.80 | 90.80 | 91.60 | 91.20 | 89.20 | 89.20 | 91.80 | 94.20 |
>    | Naive-ICL                      | 73.40 | 78.00 | 78.80 | 82.90 | 81.40 | 82.39 | 81.20 | 82.79 | 84.20 |
>
>    | #instances  of prompt (k value) | 5     | 7     | 9     | 11    | 13    | 15    | 17   | 19    | 21    | 23    | 25    |
>    | ------------------------------- | ----- | ----- | ----- | ----- | ----- | ----- | ---- | ----- | ----- | ----- | ----- |
>    | Naive-ICL                       | 73.40 | 78.80 | 81.40 | 81.20 | 84.20 | 87.60 | 89.0 | 90.40 | 90.80 | 90.60 | 89.80 |
>
>    We can observe that CoT-ER can effectively leverage the information from the provided instances and still perform well with a smaller number of instances. However, the performance of Naive-ICL drops rapidly as the number of instances decreases. Demonstrating CoT-ER has a more stable performance compared to Naive-ICL.
>
>
>
> 2. **Comment**: Given that large language models have the potential to achieve high performance on relation extraction tasks, it would be valuable for the authors to conduct a thorough error analysis to identify the reasons for any mistakes made by the model. This analysis could provide insights into the limitations of the proposed approach and suggest directions for future research.
>
>    **Reply**: Here, we address the error anlaysis issue with a **quantitative** and **qualitative** analysis. We will add more analysis and discussion in the final paper.
>
>    **(1) Quantitative Analysis**
>
>    Taking 5-Way 1-Shot as an example, the table below shows the accuracy of these two methods on different relations of the FewRel. Note that different numbers represent different relation types, and their order is based on the performance ranking of Naive-ICL. These accuracies are the average of multiple test runs conducted with 7 different random seeds.
>
>    The table below shows the experimental results on **FewRel2.0**. We can observe that CoT-ER surpasses Naive-ICL 6 times, with 4 of them showing a significant improvement of over 10%. However, in the 5th and 7th relations, CoT-ER still lags behind Naive-ICL by a few percentage points.
>
>    | Relation Type | 1     | 2        | 3        | 4     | 5     | 6         | 7     | 8         | 9         | 10       |
>    | ------------- | ----- | -------- | -------- | ----- | ----- | --------- | ----- | --------- | --------- | -------- |
>    | Naive-ICL     | 98.71 | 95.71    | 88.82    | 88.19 | 79.33 | 75.54     | 74.07 | 70.71     | 67.69     | 65.46    |
>    | CoT-ER        | 97.84 | 95.95    | 89.29    | 87.3  | 71.32 | 93.01     | 71.32 | 89.84     | 80.41     | 81.16    |
>    | Difference    | -0.87 | **0.24** | **0.47** | -0.89 | -8.01 | **17.47** | -2.75 | **19.13** | **12.72** | **15.7** |
>
>    (Definition of digital relation type: 1 inheritance type of, 2 ingredient of, 3 classified as, 4 gene found in organism, 5 is primary anatomic site of disease, 6 causative agent of, 7 biological process involves gene product, 8 gene plays role in process, 9 is normal tissue origin of disease, 10 occurs in)
>
>
>
>    The table below shows the experimental results on **FewRel1.0**. We can observe that CoT-ER surpasses Naive-ICL 10 times, with 7 of them showing a relatively high improvement. However, in the 7th and 8th relations, CoT-ER still lags behind Naive-ICL by a few percentage points.
>
>    | Relation Type | 1    | 2    | 3       | 4     | 5       | 6        | 7     | 8     | 9        | 10      | 11    | 12       | 13       | 14        | 15       | 16       |
>    | ------------- | ---- | ---- | ------- | ----- | ------- | -------- | ----- | ----- | -------- | ------- | ----- | -------- | -------- | --------- | -------- | -------- |
>    | Naive-ICL     | 100  | 100  | 99.12   | 98.63 | 98.3    | 98.13    | 97.72 | 96.64 | 96.57    | 94.15   | 92.94 | 91.01    | 85.24    | 84.63     | 83.71    | 82.47    |
>    | CoT-ER        | 100  | 100  | 99.52   | 98.07 | 100     | 98.65    | 95.51 | 89.47 | 100      | 95.25   | 91.72 | 95.66    | 89.31    | 95.98     | 86.95    | 91.39    |
>    | Difference    | 0    | 0    | **0.4** | -0.56 | **1.7** | **0.52** | -2.21 | -7.17 | **3.43** | **1.1** | -1.22 | **4.65** | **4.07** | **11.35** | **3.24** | **8.92** |
>
>    (Definition of digital relation type:1 voice type, 2 position played on team/speciality, 3 original language of film or TV show, 4 constellation, 5 military rank, 6 competition class, 7 member of, 8 spouse, 9 located in or next to body of water, 10 follows, 11 crosses, 12 sport, 13 main subject, 14 child, 15 part of, 16 mother)
>
> **(2) Qualitative Analysis**
>
> In summary, CoT-ER shows significant improvements in scenarios when Naive-ICL performs poorly. Here, we present a few cases to illustrate the reasoning process of CoT-ER in some representative scenarios. More details will be added to the revised paper. We will showcase two query instances with the digital relation types 8 (the highest difference) and 5 (the lowest difference) in the FewRel2.0 above.
>
> **(i) Positive Example**
>
> - Context:
>
>   In the hippocampus, transcriptional upregulation was observed in nogo-a(one day post-injury), mag and pirb at seven days post-injury.
>
> - Reasoning process of CoT-ER:
>
>   It can be inferred from the context that:
>
>   - **Entity "transcriptional" refers to the process of transcribing DNA into RNA**, which refers to the entity of biological process in the context.
>   - **Entity "mag" is the name of a gene**, which refers to the entity of gene in the context.
>   - According to the context, "transcriptional upregulation was observed in nogo-a (one day post-injury), mag and pirb at seven days post-injury" indicates that "mag" plays role in "transcriptional".
>
>   So, the relation between "transcriptional" and "mag" is **"gene plays role in process"**.
>
> In this case, CoT-ER can not only recognize what "transcriptional" and "mag" mean in the context but also have concept-level knowledge. And the final prediction is correct.
>
> **(ii) Negative Example**
>
> - Context:
>
>   Loss of hair in the other areas of the skin is present in the majority of cases.
>
> - Reasoning process of CoT-ER:
>
>   It can be inferred from the context that:
>
>   - **Entity "Loss of hair" is the name of a condition**, which refers to the entity of condition in the context.
>
>   - **Entity "skin" is the name of a body organ**, which refers to the entity of physical location of body part in the context.
>
>   - According to the context, "loss of hair in the other areas of the skin" indicates that "loss of hair" occurs in "skin".
>
>   So, the relation between "loss of hair" and "skin" is `"occurs in"`.
>
> In this case, CoT-ER can also recognize the entity well, and the conclusion is semantically right ("loss of hair" occurs in "skin"). However, **the final prediction is incorrect**, as the predicted relation label is "occurs in" while the ground truth label is "is primary anatomic site of disease".
>
> The possible reason is that the label "occurs in" in this dataset means "a condition occurs in a period of the lifetime". This particular label should be matched with entity pairs like "condition or disease" and "a period of a person's lifetime (such as congenital)". This issue indicates that the relation description would be a key component in such methods, but it's not included in the FewRel2.0 dataset.

---

### Official Review · Reviewer_josS · 2023-08-04

**Typos Grammar Style And Presentation Improvements:** 1. A space should be placed between t…
**Soundness:** 3

**Excitement:**

3: Ambivalent: It has merits (e.g., it reports state-of-the-art results, the idea is nice), but there are key weaknesses (e.g., it describes incremental work), and it can significantly benefit from another round of revision. However, I won't object to accepting it if my co-reviewers champion it.

**Paper Topic And Main Contributions:**

This paper focuses on the issue of few-shot relation extraction, which involves identifying relations in a query set based on only a few training examples. While previous research has mainly focused on supervised learning methods, this paper aims to address the problem using training-free approaches that utilize large language models.

The main contributions of this paper are twofold. Firstly, it explores the potential of in-context learning in few-shot relation extraction, bridging the existing gap. Secondly, it proposes a 3-step typed chain-of-thought prompts, where the first two steps involve eliciting entity type information and then utilizing this information along with the context to identify relations. The experimental results demonstrate that the inclusion of entity type information leads to improvements in the FewRel 1.0 and FewRel 2.0 validation sets.

**Questions For The Authors:**

A: Could you provide more details about the meaning of the line 279-382, which states "Due to the high cost of running the CoT-ER on the GPT-3 API, we evaluate all LLM-relevant methods by sampling 100 N-way K-shot tasks from each dataset"?

B: How significant are the improvements observed in Tab1 and Tab2 in the test results?

C: It would be intriguing to investigate the types of questions that benefit from the reasoning process.

D: Why were only the validation sets selected for Tab 1 and Tab 2? Line 346 - 355 mentions the difference in relation label descriptions between FewRel1.0 and FewRel2.0. Are these relation descriptions used in the prompts? If not, why including the test set would cause unfair comparision?

E: Could you explain in detail how the Verbalize() function in Figure 3 operates?

F: According to Figure 3, the instruction states that "the relation to choose must be in these N possible relations." Does the model always follow this instruction and select a relation from the given labels?

G: Have you conducted any error analysis on the CoT-ER? Considering the high performance of CoT-ER, it would be interesting to identify the types of cases that still cannot be solved.

**Reasons To Accept:**

This paper contributes to the field of relation extraction by providing a baseline for in-context learning and an entity-aware prompting appraoch. It empirically demonstrates that incorporating entity information into prompts leads to further performance improvement in few-shot relation extraction.

While the effectiveness of entity information in supervised relation learning has been previously shown, this paper focuses on incorporating entity type and explicit information from large-language models (LLMs) in prompting. This finding shows the importance of entity information in facilitating reasoning and strengthens the case for its inclusion in LLM-based approaches for training-free relation extraction.

**Reasons To Reject:**

The primary weaknesses of this paper include incomplete presentation of results in Tab1 and Tab2. For a fair comparison, the paper should provide results for both the validation and test sets. Additionally, the absence of significance tests to determine the statistical significance of the improvements compared to existing approaches is a limitation. This raises uncertainty regarding the significance of the reported improvements.

Moreover, the similarity retrieval process employed in this paper not only incurs higher computational costs but also prolongs the prompting process, making it non-end-to-end. This limitation should be addressed to ensure a more efficient and streamlined approach.

**Reproducibility:**

4: Could mostly reproduce the results, but there may be some variation because of sample variance or minor variations in their interpretation of the protocol or method.

**Reviewer Confidence:**

4: Quite sure. I tried to check the important points carefully. It's unlikely, though conceivable, that I missed something that should affect my ratings.

---

> ### Author Rebuttal · Authors · 2023-08-29
>
> Thank you for your valuable comments. We will explain your concerns point by point.
>
> 1. **Comment**: Could you provide more details about the meaning of the line 279-382, which states "Due to the high cost of running the CoT-ER on the GPT-3 API, we evaluate all LLM-relevant methods by sampling 100 N-way K-shot tasks from each dataset"?
>
>    **Reply**: First, due to limitations on the *text-davinci-003* API usage (such as access rate limit) and the higher cost of access ($0.02/1K tokens), we had to reduce the sampling number. Second, we have evaluated the CoT-ER and Naive-ICL **with 8 random seeds** (i.e., 8 independent sampling for the meta-task) in order to address this problem and will add these results to the final paper. The Table below shows experimental results with **mean ± standard deviation** on FewRel2.0. We can observe that **CoT-ER consistently outperforms Naive-ICL in all N-way K-shot settings with a lower standard deviation**.
>
>    | Method       | 5-Way 1-Shot | 5-Way 5-Shot | 10-Way 1-Shot | 10-Way 5-Shot |
>    | ------------ | ------------ | ------------ | ------------- | ------------- |
>    | Naive-ICL    | 80.75±1.7    | 88.19±1.56   | 68.21±1.27    | 80.92±1.05    |
>    | CoT-ER(ours) | 85.64±1.28   | 92.99±1.43   | 77.61±0.87    | 86.18±0.88    |
>
>
>
> 2. **Comment**: How significant are the improvements observed in Tab1 and Tab2 in the test results?
>    Why were only the validation sets selected for Tab 1 and Tab 2? Line 346 - 355 mentions the difference in relation label descriptions between FewRel1.0 and FewRel2.0. Are these relation descriptions used in the prompts? If not, why including the test set would cause unfair comparision?
>
>    **Reply**: Both FewRel1.0 and FewRel2.0 have released their training and validation sets for public use. However,  the downloaded test set from the benchmark lacks semantic label names for each instance (such as "sport", "member of"). Instead, it employs class label IDs like 0, 1, 2, 3, 4. It means that the prompting method **cannot be deployed on these datasets**. More details can be found on the official FewRel Github repository.
>
>    FewRel1.0 provides a description of each relation (such as "sport: sport in which the subject participates or belongs to") in a separate file, while FewRel2.0 does not. This has an impact on the **Verbalize() function** mentioned in the paper, and we will talk about it next. However, these descriptions are not used in the prompts.
>
>    The term "fairness" in the paper refers to the exact same experimental settings of a comparison between our method and baseline methods (including fully supervised methods). However, since our method cannot be experimented with on the test set, we primarily contrast its performance with baseline methods on the validation set with the same settings. The performance on the test set is only shown as a reference.
>
>
>
> 3. **Comment**: Could you explain in detail how the Verbalize() function in Figure 3 operates?
>
>    **Reply**:  We apologize for the unclear description and will update our paper accordingly.
>    To ensure the semantic coherence of the reasoning process, we devised this label verbalization technique by integrating both entities and the semantic relation label into a coherent expression. Give a simple example, if the entity "Demetrius Rhaney" and "American football" holds a relation label "sport", then the output of **Verbalize("Demetrius Rhaney", "sport", "American football")** is **"Demetrius Rhaney participates or belongs to American football"**.
>
>
>
> 4. **Comment**: According to Figure 3, the instruction states that "the relation to choose must be in these N possible relations." Does the model always follow this instruction and select a relation from the given labels?
>
>    **Reply**: Yes, due to the uncertainty of the model's output, we have to prevent it from outputting a relation that is not in the meta-task. In addition, the metrics of FewRel1.0 and FewRel2.0 also require this.
>
>
>
> 5. **Comment**: It would be intriguing to investigate the types of questions that benefit from the reasoning process. Have you conducted any error analysis on the CoT-ER? Considering the high performance of CoT-ER, it would be interesting to identify the types of cases that still cannot be solved.
>
>    **Reply**: Here, we address the error anlaysis issue with a **quantitative** and **qualitative** analysis. We will add more analysis and discussion in the final paper.
>
>    **(1) Quantitative Analysis**
>
>    Taking 5-Way 1-Shot as an example, the table below shows the accuracy of these two methods on different relations of the FewRel. Note that different numbers represent different relation types, and their order is based on the performance ranking of Naive-ICL. These accuracies are the average of multiple test runs conducted with 7 different random seeds.
>
>    The table below shows the experimental results on **FewRel2.0**. We can observe that CoT-ER surpasses Naive-ICL 6 times, with 4 of them showing a significant improvement of over 10%. However, in the 5th and 7th relations, CoT-ER still lags behind Naive-ICL by a few percentage points.
>
>    | Relation Type | 1     | 2        | 3        | 4     | 5     | 6         | 7     | 8         | 9         | 10       |
>    | ------------- | ----- | -------- | -------- | ----- | ----- | --------- | ----- | --------- | --------- | -------- |
>    | Naive-ICL     | 98.71 | 95.71    | 88.82    | 88.19 | 79.33 | 75.54     | 74.07 | 70.71     | 67.69     | 65.46    |
>    | CoT-ER        | 97.84 | 95.95    | 89.29    | 87.3  | 71.32 | 93.01     | 71.32 | 89.84     | 80.41     | 81.16    |
>    | Difference    | -0.87 | **0.24** | **0.47** | -0.89 | -8.01 | **17.47** | -2.75 | **19.13** | **12.72** | **15.7** |
>
>    (Definition of digital relation type: 1 inheritance type of, 2 ingredient of, 3 classified as, 4 gene found in organism, 5 is primary anatomic site of disease, 6 causative agent of, 7 biological process involves gene product, 8 gene plays role in process, 9 is normal tissue origin of disease, 10 occurs in)
>
>
>
>    The table below shows the experimental results on **FewRel1.0**. We can observe that CoT-ER surpasses Naive-ICL 10 times, with 7 of them showing a relatively high improvement. However, in the 7th and 8th relations, CoT-ER still lags behind Naive-ICL by a few percentage points.
>
>    | Relation Type | 1    | 2    | 3       | 4     | 5       | 6        | 7     | 8     | 9        | 10      | 11    | 12       | 13       | 14        | 15       | 16       |
>    | ------------- | ---- | ---- | ------- | ----- | ------- | -------- | ----- | ----- | -------- | ------- | ----- | -------- | -------- | --------- | -------- | -------- |
>    | Naive-ICL     | 100  | 100  | 99.12   | 98.63 | 98.3    | 98.13    | 97.72 | 96.64 | 96.57    | 94.15   | 92.94 | 91.01    | 85.24    | 84.63     | 83.71    | 82.47    |
>    | CoT-ER        | 100  | 100  | 99.52   | 98.07 | 100     | 98.65    | 95.51 | 89.47 | 100      | 95.25   | 91.72 | 95.66    | 89.31    | 95.98     | 86.95    | 91.39    |
>    | Difference    | 0    | 0    | **0.4** | -0.56 | **1.7** | **0.52** | -2.21 | -7.17 | **3.43** | **1.1** | -1.22 | **4.65** | **4.07** | **11.35** | **3.24** | **8.92** |
>
>    (Definition of digital relation type: 1 voice type, 2 position played on team/speciality, 3 original language of film or TV show, 4 constellation, 5 military rank, 6 competition class, 7 member of, 8 spouse, 9 located in or next to body of water, 10 follows, 11 crosses, 12 sport, 13 main subject, 14 child, 15 part of, 16 mother)
>
>    **(2) Qualitative Analysis**
>
>    In summary, CoT-ER shows significant improvements in scenarios when Naive-ICL performs poorly. Here, we present a few cases to illustrate the reasoning process of CoT-ER in some representative scenarios. More details will be added to the revised paper. We will showcase two query instances with the digital relation types 8 (the highest difference) and 5 (the lowest difference) in the FewRel2.0 above.
>
>    **(i) Positive Example**
>
>    - Context:
>
>      In the hippocampus, transcriptional upregulation was observed in nogo-a(one day post-injury), mag and pirb at seven days post-injury.
>
>    - Reasoning process of CoT-ER:
>
>      It can be inferred from the context that:
>
>      - **Entity "transcriptional" refers to the process of transcribing DNA into RNA**, which refers to the entity of biological process in the context.
>      - **Entity "mag" is the name of a gene**, which refers to the entity of gene in the context.
>      - According to the context, "transcriptional upregulation was observed in nogo-a (one day post-injury), mag and pirb at seven days post-injury" indicates that "mag" plays role in "transcriptional".
>
>      So, the relation between "transcriptional" and "mag" is **"gene plays role in process"**.
>
>    In this case, CoT-ER can not only recognize what "transcriptional" and "mag" mean in the context but also have concept-level knowledge. And the final prediction is **correct**.
>
>    **(ii) Negative Example**
>
>    - Context:
>
>      Loss of hair in the other areas of the skin is present in the majority of cases.
>
>    - Reasoning process of CoT-ER:
>
>      It can be inferred from the context that:
>
>      - **Entity "Loss of hair" is the name of a condition**, which refers to the entity of condition in the context.
>
>      - **Entity "skin" is the name of a body organ**, which refers to the entity of physical location of body part in the context.
>
>      - According to the context, "loss of hair in the other areas of the skin" indicates that "loss of hair" occurs in "skin".
>
>      So, the relation between "loss of hair" and "skin" is `"occurs in"`.
>
>    In this case, CoT-ER can also recognize the entity well, and the conclusion is semantically right ("loss of hair" occurs in "skin"). However, **the final prediction is incorrect**, as the predicted relation label is "occurs in" while the ground truth label is "is primary anatomic site of disease".
>
>    The possible reason is that the label "occurs in" in this dataset means "a condition occurs in a period of the lifetime". This particular label should be matched with entity pairs like "condition or disease" and "a period of a person's lifetime (such as congenital)". This issue indicates that the relation description would be a key component in such methods, but it's not included in the FewRel2.0 dataset.
>
>
>
> 6. **Comment**: Moreover, the similarity retrieval process employed in this paper not only incurs higher computational costs but also prolongs the prompting process, making it non-end-to-end. This limitation should be addressed to ensure a more efficient and streamlined approach.
>
>    **Reply**:
>
>    The computational costs of the retrieval module are not as high as commonly expected, because the scope of retrieval is limited to the support set within the meta-task. Additionally, the model used for computing semantic embedding vectors can be easily converted to other models that can be deployed locally, such as SimCSE (Gao, et al., EMNLP 2021). Consequently, **the additional computational cost brought by the retrieval module is at a relatively lower level** compared to getting responses from the LLM through network-accessed APIs.
>
>    We introduced this mainly considering **the inability to incorporate the entire support set instances into the prompt**, due to the limited context window of text-davinci-003 (4097 tokens). A relevant former study suggests that selecting few-shot demonstrations based on similarity yields improvements in in-context learning (Liu et al., DeeLIO 2022), while these relevant studies in order to select semantically close instances from a large training dataset, which is not a true few-shot learning (Perez et al., NIPS 2021). Many recent efforts are devoted to expanding the length of the context window. We believe that this issue will be addressed correspondingly in the near future.
>
>    Thanks again for the comments, we will provide a clearer elaboration on this in the revised paper.
>
>    References:
>
>    - What Makes Good In-Context Examples for GPT-3? (Liu et al., DeeLIO 2022)
>
>    - True few-shot learning with language models (Perez et al., NIPS 2021)
>
>    - SimCSE: Simple Contrastive Learning of Sentence Embeddings (Gao et al., EMNLP 2021)
>
>
>
> 7. **Comment**: Additionally, the absence of significance tests to determine the statistical significance of the improvements compared to existing approaches is a limitation. This raises uncertainty regarding the significance of the reported improvements.
>
>    **Reply**: Thanks a lot for your advice, We have conducted the experiment and more details will be updated to the final paper.
>
>    |      | 5-Way 1-Shot   | 5-Way 5-Shot   | 10-Way 1-Shot  | 10-Way 5-Shot  |
>    | ---- | -------------- | -------------- | -------------- | -------------- |
>    | 1    | $1.1×10^{-3}$  | $3.74×10^{-2}$ | $1.77×10^{-8}$ | $3.66×10^{-3}$ |
>    | 2    | $1.1×10^{-2}$  | $8.04×10^{-3}$ | $1.41×10^{-4}$ | $5.08×10^{-5}$ |
>    | 3    | $4.00×10^{-2}$ | $1.30×10^{-2}$ | $1.55×10^{-8}$ | $1.36×10^{-2}$ |
>    | 4    | $5.26×10^{-3}$ | $8.67×10^{-3}$ | $8.75×10^{-6}$ | $2.68×10^{-5}$ |
>    | 5    | $5.99×10^{-3}$ | $2.35×10^{-3}$ | $8.25×10^{-9}$ | $5.12×10^{-3}$ |
>
>    This table displays the **p-value** between Naivel-ICL and CoT-ER on FewRel2.0 across 5 different random seeds, which is obtained by sampling for each meta-test in all meta-tasks. **All the p-values are below 0.05, indicating significant differences between the two**.

---

### Official Review · Reviewer_cCve · 2023-08-05

**Soundness:** 4

**Excitement:**

3: Ambivalent: It has merits (e.g., it reports state-of-the-art results, the idea is nice), but there are key weaknesses (e.g., it describes incremental work), and it can significantly benefit from another round of revision. However, I won't object to accepting it if my co-reviewers champion it.

**Paper Topic And Main Contributions:**

In this paper, the authors propose an approach to effectively prompt LLMs for few-shot relation extraction. The approach - CoT-ER, utilizes in-context learning and chain-of-thought prompting techniques to create a three step process for relation extraction. First, a reasoning module is used to generate/manually extract concepts related to the head, tail, and relation, along with the relevant text (evidence) of the samples present in the few-shot samples support set. In the second step, a subset of the few-shot demonstrations are selected, using kNN of embeddings from the query sample. This step ensures that the selected subset do not exceed the token length limit of the LLM. The final step involves inferring the relation label for the query sample by concatenating the task description and the selected few-shot demonstrations with the query text. Experiments demonstrate that CoT-ER improves upon various baselines over a subset of test samples chosen from the FewRel 1.0 and FewRel 2.0 datasets. Further, ablations show the importance of the first two steps, and a case study of a single sample demonstrates the different steps involved in the CoT-ER approach.

**Contributions:**

- An new approach for effectively prompting  LLMs for the few-shot relation extraction task providing competitive performance against fully fine tuned baselines.
- A comprehensive comparison of fully-trained and alternative prompting approaches for few-shot relation extraction.

**Questions For The Authors:**

**Question A:**  In what scenarios does CoT-ER provide an improvement over Naive-ICL?

**Question B:** How is the number of demonstrations selected from the entire few-shot samples determined?

**Reasons To Accept:**

- A well-thought out and implemented approach that combines various ideas from prompting LLMs to for effective few-shot relation extraction.
- Explanations for the setups involved in alternative prompting approaches few-shot  relation extraction.

**Reasons To Reject:**

- No error analysis - The naive in context learning  (Naive-ICL) baseline results are quite competitive with the best results from CoT-ER, the proposed approach. Some error analysis may help justify CoT-ER as an approach over the baseline.
- Unclear selection of few-shot subset: $M$, the number of few-shot instances selected for reasoning vary a lot across the different N-way K-shot setups (5,1 0, 13, 15, 25, 40), and it is unclear how the selected numbers contribute towards the final performance.
- Evaluation on a sampled subset of the full data: Only 100 samples are used from each dataset for evaluation. Thus, the results may not hold up when evaluated on the full test datasets.

**Reproducibility:**

3: Could reproduce the results with some difficulty. The settings of parameters are underspecified or subjectively determined; the training/evaluation data are not widely available.

**Reviewer Confidence:**

3: Pretty sure, but there's a chance I missed something. Although I have a good feel for this area in general, I did not carefully check the paper's details, e.g., the math, experimental design, or novelty.

---

> ### Author Rebuttal · Authors · 2023-08-29
>
> Thank you for your valuable comments. We will explain your concerns point by point.
>
> 1. **Comment**: Only 100 samples are used from each dataset for evaluation. Thus, the results may not hold up when evaluated on the full test datasets.
>
>    **Reply**: Thanks for your insightful suggestions. It is worth mentioning that 100 samples here mean 100  meta-tasks, which are randomly sampled from instances in the dataset. Each N-way K-shot meta-task contains N queries for each relation class, which means there are $100 \times N \times (K+1)$ instances in total. Due to limitations on the **text-davinci-003** API usage (such as access rate limit) and the higher cost of access ($0.02/1K tokens), we had to reduce the sampling number.
>
>    In addition, we have evaluated the CoT-ER and Naive-ICL **with 8 random seeds** (i.e., 8 independently sampling for the meta-task) in order to address this problem and will add these results to the final paper. The table below shows experimental results with **mean ± standard deviation** on FewRel2.0. We can observe that **CoT-ER consistently outperforms Naive-ICL in all N-way K-shot settings with a lower standard deviation**.
>
>    | Method       | 5-Way 1-Shot | 5-Way 5-Shot | 10-Way 1-Shot | 10-Way 5-Shot |
>    | ------------ | ------------ | ------------ | ------------- | ------------- |
>    | Naive-ICL    | 80.75±1.7    | 88.19±1.56   | 68.21±1.27    | 80.92±1.05    |
>    | CoT-ER(ours) | 85.64±1.28   | 92.99±1.43   | 77.61±0.87    | 86.18±0.88    |
>
>
>
> 2. **Comment**: No error analysis - The naive in context learning (Naive-ICL) baseline results are quite competitive with the best results from CoT-ER, the proposed approach. Some error analysis may help justify CoT-ER as an approach over the baseline.
>
>    Question A: In what scenarios does CoT-ER provide an improvement over Naive-ICL?
>
>    **Reply**: Here, we address the error analysis issue with a **quantitative** and **qualitative** analysis. We will add more analysis and discussion in the final paper.
>
>    **(1) Quantitative Analysis**
>
>    Taking 5-Way 1-Shot as an example, the table below shows the accuracy of these two methods on different relations of the FewRel. Note that different numbers represent different relation types, and their order is based on the performance ranking of Naive-ICL. These accuracies are the average of multiple test runs conducted with 7 different random seeds.
>
>    The table below shows the experimental results on **FewRel2.0**. We can observe that CoT-ER surpasses Naive-ICL 6 times, with 4 of them showing a significant improvement of over 10%. However, in the 5th and 7th relations, CoT-ER still lags behind Naive-ICL by a few percentage points.
>
>    | Relation Type | 1     | 2        | 3        | 4     | 5     | 6         | 7     | 8         | 9         | 10       |
>    | ------------- | ----- | -------- | -------- | ----- | ----- | --------- | ----- | --------- | --------- | -------- |
>    | Naive-ICL     | 98.71 | 95.71    | 88.82    | 88.19 | 79.33 | 75.54     | 74.07 | 70.71     | 67.69     | 65.46    |
>    | CoT-ER        | 97.84 | 95.95    | 89.29    | 87.3  | 71.32 | 93.01     | 71.32 | 89.84     | 80.41     | 81.16    |
>    | Difference    | -0.87 | **0.24** | **0.47** | -0.89 | -8.01 | **17.47** | -2.75 | **19.13** | **12.72** | **15.7** |
>
>    (Definition of digital relation type: 1 inheritance type of, 2 ingredient of, 3 classified as, 4 gene found in organism, 5 is primary anatomic site of disease, 6 causative agent of, 7 biological process involves gene product, 8 gene plays role in process, 9 is normal tissue origin of disease, 10 occurs in)
>
>
>
>    The table below shows the experimental results on **FewRel1.0**. We can observe that CoT-ER surpasses Naive-ICL 10 times, with 7 of them showing a relatively high improvement. However, in the 7th and 8th relations, CoT-ER still lags behind Naive-ICL by a few percentage points.
>
>    | Relation Type | 1    | 2    | 3       | 4     | 5       | 6        | 7     | 8     | 9        | 10      | 11    | 12       | 13       | 14        | 15       | 16       |
>    | ------------- | ---- | ---- | ------- | ----- | ------- | -------- | ----- | ----- | -------- | ------- | ----- | -------- | -------- | --------- | -------- | -------- |
>    | Naive-ICL     | 100  | 100  | 99.12   | 98.63 | 98.3    | 98.13    | 97.72 | 96.64 | 96.57    | 94.15   | 92.94 | 91.01    | 85.24    | 84.63     | 83.71    | 82.47    |
>    | CoT-ER        | 100  | 100  | 99.52   | 98.07 | 100     | 98.65    | 95.51 | 89.47 | 100      | 95.25   | 91.72 | 95.66    | 89.31    | 95.98     | 86.95    | 91.39    |
>    | Difference    | 0    | 0    | **0.4** | -0.56 | **1.7** | **0.52** | -2.21 | -7.17 | **3.43** | **1.1** | -1.22 | **4.65** | **4.07** | **11.35** | **3.24** | **8.92** |
>
>    (Definition of digital relation type:1 voice type, 2 position played on team/speciality, 3 original language of film or TV show, 4 constellation, 5 military rank, 6 competition class, 7 member of, 8 spouse, 9 located in or next to body of water, 10 follows, 11 crosses, 12 sport, 13 main subject, 14 child, 15 part of, 16 mother)
>
>
>
>    **(2) Qualitative Analysis**
>
>    In summary, CoT-ER shows significant improvements in scenarios when Naive-ICL performs poorly. Here, we present a few cases to illustrate the reasoning process of CoT-ER in some representative scenarios. More details will be added to the revised paper. We will showcase two query instances with the digital relation types 8 (the highest difference) and 5 (the lowest difference) in the FewRel2.0 above.
>
>    **(i) Positive Example**
>
>       - Context:
>
>         In the hippocampus, transcriptional upregulation was observed in nogo-a(one day post-injury), mag and pirb at seven days post-injury.
>
>       - Reasoning process of CoT-ER:
>
>         It can be inferred from the context that:
>
>         - **Entity "transcriptional" refers to the process of transcribing DNA into RNA**, which refers to the entity of biological process in the context.
>         - **Entity "mag" is the name of a gene**, which refers to the entity of gene in the context.
>         - According to the context, "transcriptional upregulation was observed in nogo-a (one day post-injury), mag and pirb at seven days post-injury" indicates that "mag" plays role in "transcriptional".
>
>         So, the relation between "transcriptional" and "mag" is **"gene plays role in process"**.
>
>    In this case, CoT-ER can not only recognize what "transcriptional" and "mag" mean in the context but also have concept-level knowledge. And the final prediction is correct.
>
>
>
>    **(ii) Negative Example**
>
>       - Context:
>
>         Loss of hair in the other areas of the skin is present in the majority of cases.
>
>       - Reasoning process of CoT-ER:
>
>         It can be inferred from the context that:
>
>         - **Entity "Loss of hair" is the name of a condition**, which refers to the entity of condition in the context.
>
>         - **Entity "skin" is the name of a body organ**, which refers to the entity of physical location of body part in the context.
>
>         - According to the context, "loss of hair in the other areas of the skin" indicates that "loss of hair" occurs in "skin".
>
>         So, the relation between "loss of hair" and "skin" is `"occurs in"`.
>
>    In this case, CoT-ER effectively recognizes the entity and arrives at a semantically accurate conclusion ("loss of hair" occurs in "skin"). However, **the final prediction is incorrect**, as the predicted relation label is "occurs in" while the ground truth label is "is primary anatomic site of disease".
>
>    The possible reason is that the label "occurs in" in this dataset means "a condition occurs in a period of the lifetime". This particular label should be matched with entity pairs like "condition or disease" and "a period of a person's lifetime (such as congenital)". This issue indicates that the relation description would be a key component in such methods, but it's not included in the FewRel2.0 dataset.
>
>
>
> 3. **Comment**: Question B: How is the number of demonstrations selected from the entire few-shot samples determined?
>
>    **Reply**: Sorry for the unclear description. We will clarify this point in the final paper. An N-way K-shot meta-task contains $N\times K$ support instances in total, while the **text-davinci-003** which has a context window of 4097 tokens could not hold all support instances. Intuitively, we aim to **provide as much information as possible for LLM**, so we follow the principle of filling the context window to select the number of demonstrations as much as possible.
>
>
>
> 4. **Comment**: The number of few-shot instances selected for reasoning vary a lot across the different N-way K-shot setups (5,1 0, 13, 15, 25, 40), and it is unclear how the selected numbers contribute towards the final performance.
>
>    **Reply**: There are two main reasons for this. First, the support set instances vary for each meta-task ($N\times K$ instances for each N-way K-shot task). Second, the number of tokens in each instance varies depending on the method employed (~300 in CoT-ER and ~80 in Naive-ICL), but they are all determined by the intuition mentioned above.
>
>    | #few-shot instances | 5     | 6     | 7     | 8     | 9     | 10    | 11    | 12    | 13    |
>    | ------------------- | ----- | ----- | ----- | ----- | ----- | ----- | ----- | ----- | ----- |
>    | CoT-ER              | 88.40 | 88.80 | 90.80 | 91.60 | 91.20 | 89.20 | 89.20 | 91.80 | 94.20 |
>    | Naive-ICL           | 73.40 | 78.00 | 78.80 | 82.90 | 81.40 | 82.39 | 81.20 | 82.79 | 84.20 |
>
>    | #few-shot instances | 5     | 7     | 9     | 11    | 13    | 15    | 17   | 19    | 21    | 23    | 25    |
>    | ------------------- | ----- | ----- | ----- | ----- | ----- | ----- | ---- | ----- | ----- | ----- | ----- |
>    | Naive-ICL           | 73.40 | 78.80 | 81.40 | 81.20 | 84.20 | 87.60 | 89.0 | 90.40 | 90.80 | 90.60 | 89.80 |
>
>    We can observe that CoT-ER can effectively leverage the information from the provided instances and still perform well with a smaller number of instances. However, the performance of Naive-ICL drops rapidly as the number of instances decreases. Demonstrating that CoT-ER has a more stable performance compared to Naive-ICL.

---

### Meta-Review · Area_Chair_Nxvf · 2023-09-25

**Recommendation:** 3

**Metareview:**

This paper tackles the challenge of few-shot relation extraction by employing training-free methods, in contrast to traditional supervised learning. It achieves two key contributions: (1) it investigates the effectiveness of in-context learning for this problem; (2) it introduces a 3-step approach using typed chain-of-thought prompts. This method starts by extracting entity type information and then uses it in conjunction with the context to identify relations. Experimental results show that including entity type information enhances performance on the FewRel 1.0 and FewRel 2.0 validation sets. Reviewers unanimously agreed on the good soundness of the work (one reviewer also increased soundness score after rebuttal), and also shared several areas of improvement. First, there's a lack of error analysis, which could help justify the proposed CoT-ER approach over the baseline. Second, the selection of few-shot instances for reasoning varies widely across setups, and its contribution to performance is unclear. Third, only 100 samples are used from each dataset for evaluation. Thus, the results may not hold up when evaluated on the full test datasets. Finally, the paper does not provide a detailed analysis of the impact of the k value on the performance of the proposed method. k-NN search is a key component, a more thorough analysis of the impact of the k value on the experimental results is needed.

---

### Decision · Program_Chairs · 2023-10-07

**Decision:**

Accept-Findings

**Comment:**

This paper tackles the challenge of few-shot relation extraction by employing training-free methods, in contrast to traditional supervised learning. It achieves two key contributions: (1) it investigates the effectiveness of in-context learning for this problem; (2) it introduces a 3-step approach using typed chain-of-thought prompts. This method starts by extracting entity type information and then uses it in conjunction with the context to identify relations. Experimental results show that including entity type information enhances performance on the FewRel 1.0 and FewRel 2.0 validation sets. Reviewers unanimously agreed on the good soundness of the work (one reviewer also increased soundness score after rebuttal), and also shared several areas of improvement. First, there's a lack of error analysis, which could help justify the proposed CoT-ER approach over the baseline. Second, the selection of few-shot instances for reasoning varies widely across setups, and its contribution to performance is unclear. Third, only 100 samples are used from each dataset for evaluation. Thus, the results may not hold up when evaluated on the full test datasets. Finally, the paper does not provide a detailed analysis of the impact of the k value on the performance of the proposed method. k-NN search is a key component, a more thorough analysis of the impact of the k value on the experimental results is needed.